# To Transfer or Not to Transfer:
# Suppressing Concepts from Source Representations

**Vijay Sadashivaiah**                                    *sadasv2@rpi.edu*
*Department of Computer Science, Rensselaer Polytechnic Institute*

**Keerthiram Murugesan**                    *keerthiram.murugesan@ibm.com*
*IBM Research, Yorktown Heights*

**Ronny Luss**                                        *rluss@us.ibm.com*
*IBM Research, Yorktown Heights*

**Pin-Yu Chen**                                  *pin-yu.chen@ibm.com*
*IBM Research, Yorktown Heights*

**Chris Sims**                                        *simsc3@rpi.edu*
*Department of Cognitive Science, Rensselaer Polytechnic Institute*

**James Hendler**                                  *hendler@cs.rpi.edu*
*Department of Computer Science, Rensselaer Polytechnic Institute*

**Amit Dhurandhar**                              *adhuran@us.ibm.com*
*IBM Research, Yorktown Heights*

**Reviewed on OpenReview:** *https://openreview.net/forum?id=BNP4MxzDEI*

## Abstract

With the proliferation of large pre-trained models in various domains, transfer learning has gained prominence where intermediate representations from these models can be leveraged to train better (target) task-specific models, with possibly limited labeled data. Although transfer learning can be beneficial in many applications, it can transfer undesirable information to target tasks that may severely curtail its performance in the target domain or raise ethical concerns related to privacy and/or fairness. In this paper, we propose a novel approach for suppressing the transfer of user-determined semantic concepts (viz. color, glasses, etc.) in intermediate source representations to target tasks without retraining the source model which can otherwise be expensive or even infeasible. Notably, we tackle a bigger challenge in the input data as a given intermediate source representation is biased towards the source task, thus possibly further entangling the desired concepts. We evaluate our approach qualitatively and quantitatively in the visual domain showcasing its efficacy for classification and generative source models. Finally, we provide a concept selection approach that automatically suppresses the undesirable concepts.

## 1 Introduction

Deep neural networks (DNN) have achieved unprecedented performance in various computer vision and natural language (NLP) problems such as image classification (Sun et al., 2017; Mahajan et al., 2018), object detection (Girshick, 2015; Ren et al., 2015), segmentation (Long et al., 2015; He et al., 2017), question answering (Min et al., 2017; Chung et al., 2017), and machine translation (Zoph et al., 2016; Wang et al., 2018) etc. One of their strengths is learning task-specific hidden representations rather than relying on predefined image features. In an ideal scenario, there are abundant labeled training samples to learn a good

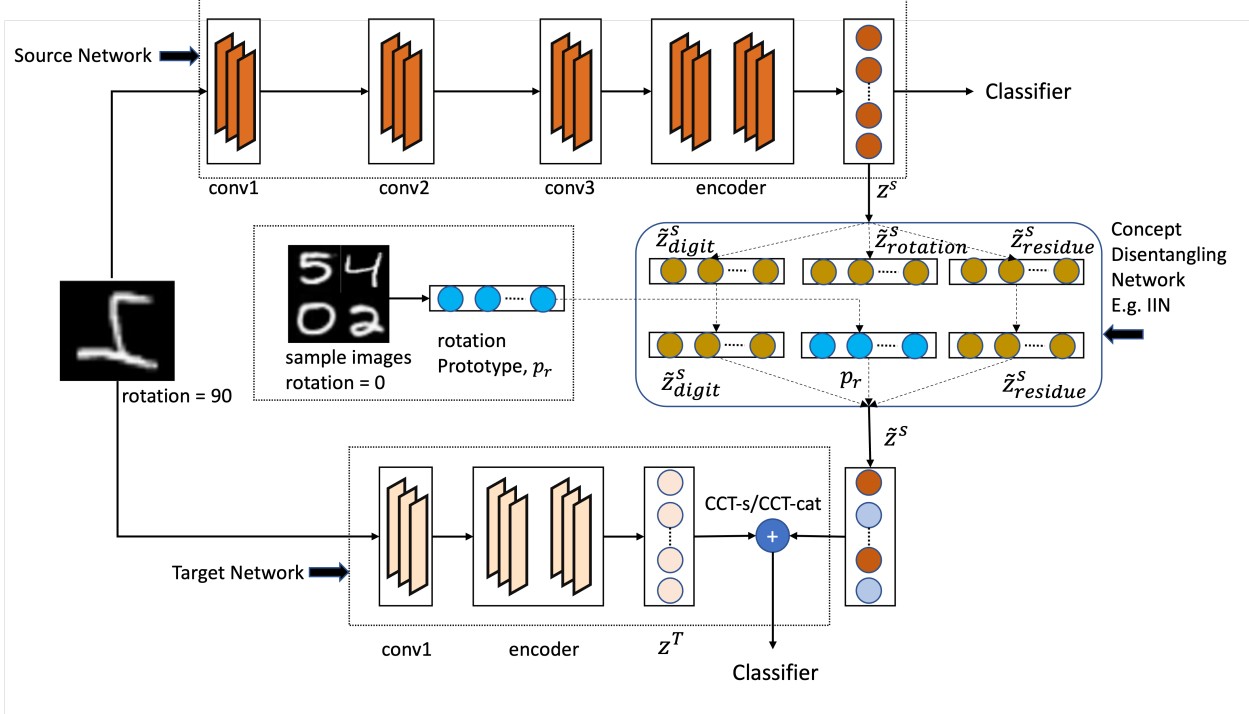

Figure 1: Illustration of our proposed approach on rotated-MNIST dataset. We show how the *rotation* concept is suppressed from the image before transferring it to the target task. First, hidden representation $z^s$ from a pre-trained source network is factorized using a concept disentangling network (CDN) into $z^s_k$, $k \in K = \{digit, rotation, residue\}$. To suppress the *rotation* concept, the factor for $z^s_{rotation}$ is set to a rotation *prototype*, $p_r$ generated using a few sample images with no rotation. Now the CDN is used to invert the modified factors to $\tilde{z}^s$ which can either be directly used *Controllable Concept Transfer-source (CCT-s)* or concatenated with the target representation $z^T$ *Controllable Concept Transfer-concatenate (CCT-cat)* to train the target task.

hidden representation. However, collecting sufficient training data is often expensive, time-consuming, or unrealistic. In such scenarios, transfer learning (Pan & Yang, 2009) has emerged as one of the promising learning paradigms. Transfer learning utilizes knowledge from information-rich source tasks to learn a specific (often information-poor) target task.

One of the most widely used approaches for transfer learning is *fine-tuning* (Sharif Razavian et al., 2014) where the target DNN being trained is initialized with the weights of a source DNN that has been pre-trained on a large dataset from a related task. Another popular approach involves combining (matching) the hidden representation (output gradient) of the target model with that of the source model (Jang et al., 2019; Li et al., 2018; Murugesan et al., 2022). These approaches are extensively used in improving prediction performance and robustness of many vision and NLP tasks (Hendrycks et al., 2019; Devlin et al., 2018) while reducing training time and resources. This effectiveness is partially attributed to the capability of DNNs to repurpose features and concepts for new tasks, as discussed by Neyshabur et al. (2020).

In the realm of image classification, for example, *visual concepts* such as shapes, colors, or textures learned while distinguishing between cats and dogs might be transferred to differentiate between tigers and wolves in a new task. While concept transfer can enhance the efficiency of models on related tasks, it could propagate undesirable *concepts* encoded in source models to downstream tasks. For example, a source model, trained to classify cats vs dogs, with most cat images in gray-scale and dog images in color, could incorrectly associate the concept of *color* to the images of the dog and pass this biased knowledge to downstream tasks. In real-world applications, this could have serious consequences. Among several examples, Steed & Caliskan (2021) showed that embeddings extracted from pre-trained image models exhibit racial and gender bias that they learn from training datasets.

While there are several approaches to *mitigate* the impact of unintended knowledge transfer in target models, ranging from data augmentation that balances target datasets (Park et al., 2018; Dixon et al., 2018) to adversarial training for generating robust hidden representations against certain spurious concepts (Zhang et al., 2018; Wang et al., 2020; Fan et al., 2021), suppressing undesirable information in the intermediate representation of the source model has been largely unexplored. Typically, large pre-trained (source) models are learned with imbalanced/biased data, and retraining these models to remove undesirable concepts might not be ideal. For one, it is expensive to retrain the source model, and access to the data used in source model pretraining could be limited. Additionally, one can suppress the *visual concepts* in the input data (i.e., masking pixels in input image), but *latent concepts* such as color, rotation, smiling, etc cannot be blocked without affecting other concepts. Our work takes a novel concept-based knowledge transfer approach where we address the following question:

*How can we most effectively **control** the intermediate representation of a source model by suppressing a specific concept while keeping other concepts (largely) intact before transferring to downstream tasks?*

Toward this goal, we propose a transfer learning method, Controllable Concept Transfer (CCT), to suppress the undesirable concepts (*visual and latent*) in the hidden representation of the source model before transferring them to a downstream task.

We introduce a novel method that utilizes a Concept Disentangling Network (CDN) to disentangle the concepts within the intermediate source representation and selectively transmit or suppress them for the target task. To the best of our knowledge, this disentanglement process has not been previously explored in the context of transfer learning. Our approach for CDN adapts recent works in interpretable disentanglement of images such as Invertible Interpretable Network (IIN) (Esser et al., 2020), Concept Activation Vectors (Kim et al., 2018; Zhou et al., 2018; Chen et al., 2020). We propose two transfer learning settings and demonstrate our approach to image classification tasks using three real-world datasets. Our system successfully suppresses concepts from the source intermediate representation in both settings. In addition, we propose a mutual information-based metric based on Belghazi et al. (2018) to quantify the measure of concept suppression. Figure 1 illustrates our approach to suppress the concept of *rotation* from the source model using CDN explained later. We evaluate the performance of controllable concept transfer both qualitatively and quantitatively. Our qualitative analysis presents decoded images with different concepts suppressed.

## 2  Related Work

**Transfer Learning** from a large pre-trained source model is a well-known approach to learning target tasks with limited labels (Pan & Yang, 2009). One of the most common transfer learning techniques is finetuning a pre-trained source model (Sharif Razavian et al., 2014), where network layers from the source model are frozen, and a new classifier head is trained for the target task. Recent works align the source and target features to transfer relevant knowledge - either by matching network weights (Xuhong et al., 2018; Jang et al., 2019), attention maps (Li et al., 2018; Zagoruyko & Komodakis, 2016), Jacobians (Srinivas & Fleuret, 2018) or model reprogramming (Chen, 2022). Another line of work uses the source model to better guide the target network by transferring feature maps automatically to improve the target task performance (Murugesan et al., 2022). In all the above methods, the transferred knowledge is typically not interpretable. To understand what knowledge is being transferred from source and target networks, a few methods use attention maps to visualize the key features from the source model useful for the target task (Murugesan et al., 2022; Jang et al., 2019) or a series of experimental analyses on the finetuned target model to study the importance of transferred knowledge (Neyshabur et al., 2020). In this paper, we take a different approach to transfer learning and propose a principled way of controlling what semantically meaningful *concepts* can be transferred from the source model to the target task.

With an increasing interest in **Model Interpretability**, several approaches have been proposed to understand the inner workings of deep neural classifiers, specifically through human understandable high-level concepts as activation vectors (Kim et al., 2018; Zhou et al., 2018; Chen et al., 2020), or individual neurons (Erhan et al., 2009; Olah et al., 2017; Zeiler & Fergus, 2014; Bau et al., 2017). However, the representation

of the semantic concepts is distributed across the hidden layers of the network (Fong & Vedaldi, 2018), and none of these methods can (confidently) claim that the features (i.e., neurons) identified from intermediate representations are associated *only* with the specific concept and are largely independent of other concepts (Montavon et al., 2017; Yosinski et al., 2015). A related line of work trains the models that explicitly encode concepts in their intermediate representations (Koh et al., 2020; Chen et al., 2020; Losch et al., 2019). However, this approach alters the network architecture and typically deteriorates overall performance (Zhou et al., 2016). Unlike these works, we propose a novel method to transfer learning by blocking or allowing the relevant concepts from the source model to the target model for better interpretability.

Unlike DNNs, **Generative Models** are trained to produce images from samples of a specific distribution. Variational auto-encoders (Kumar et al., 2018; Higgins et al., 2017) reconstruct images from a representation whose marginal distribution is matched to a standard normal distribution. Generative Adversarial Networks (GAN) (Goodfellow et al., 2020; Hoang et al., 2018) map samples from a standard normal distribution to realistic images as judged by a discriminator. While these approaches are invertible, they are not interpretable, are limited to representations with a linear structure, and cannot be applied to arbitrary representations from a source network. This motivates our choice for adapting frameworks such as Invertible Neural Networks (Dinh et al., 2014; Jacobsen et al., 2018; Kingma & Dhariwal, 2018; Esser et al., 2020) for our setup as they can identify disentangled concepts for interpretability, invert them and map them back to relevant features in the intermediate representations of the source model. StyleGAN (Karras et al., 2020) is a generative model that generates images with different "styles". However, this is inappropriate for our setup since latent directional vectors need to be known for concepts one wishes to manipulate. Although these are known for certain concepts in specific datasets (viz., age and eye ratio in the FFHQ dataset), they are not readily available for arbitrary ones that one might want to manipulate. Moreover, finding the latent representation for a given real image, which would be required for the StyleGAN, is still a topic of active research (Li et al., 2022) as the reconstructions from these patients can be imperfect.

Most state-of-the-art **disentangled representation** learning methods are based on the framework of variational auto-encoders (VAE) (Kingma & Welling, 2013; Rezende et al., 2014). However, these works do not enforce any structure on the latent space and many regularization techniques have then been proposed (Kumar et al., 2018; Higgins et al., 2017; Eastwood & Williams, 2018). Among works that enforce structure are those that learn structural causal models (Shen et al., 2022; Huang et al., 2023; Zhang et al., 2023), but they are used primarily for discriminative tasks. Xiao et al. (2020) takes a different approach to disentangling within contrastive learning literature by embedding different semantic concepts in latent spaces, where each space is invariant to one semantic concept. A key differentiator between Xiao et al. (2020) and our approach is that we propose to suppress desired semantic concepts while keeping other concepts (largely) intact. On the contrary, their approach can transfer embeddings of one/few/all concepts to a downstream task. This is a limitation, considering that it is impossible to pre-define embedding space for all possible semantic concepts. Lastly, data augmentation (Mitrovic et al., 2021; von Kügelgen et al., 2021) can also be viewed as an approach to mitigating features but generally assumes knowledge of the data generating process.

## 3  Controllable Concept Transfer Method

It has been widely observed that machine learning models learn context-specific correlations in datasets. For example, a model trained to classify different indoor scenes would learn to associate the presence of a *bed* to the output class of "bedroom" vs *couch* to the output class of "living-room". However, spurious associations in the transferred knowledge could hinder the performance of a target task. For instance, an *accent chair* could be exclusively associated with "living-room" and when the model encounters a novel environment where it is present in "bedroom", the source representation could be biased to classify the input as "living-room". A transfer learning method to modify the source intermediate representation to suppress a certain *concept* would prove useful in such situations. The canonical way to suppress a certain concept from hidden representation would be to retrain the model with new images that satisfy the desired constraint. However, predicting how the retraining affects other related concepts is hard. Our work aims to tackle this problem *directly* and modify the hidden representation of the source model in a targeted manner. For instance, in our earlier example, we would ideally want to suppress the concept of *accent chair* without affecting

other concepts in the hidden representation of the source model before transferring it to the downstream task. Next, we provide a brief overview of the Concept Disentangling Network based on recent work on interpretable disentanglement of input representations by Esser et al. (2020), after which we detail our work. This paper uses the terms "intermediate representation" and "hidden representation" interchangeably.

### 3.1 Background: Disentangling semantic concepts in intermediate representation

Let $f$ be the given neural network with $L$ layers that maps the input image $x \in \mathbb{R}^{h \times w \times c}$[1] through a series of hidden layers to the final output $f(x)$. Often, intermediate representation $E(x) \in \mathbb{R}^{H \times W \times C}$[2] at a hidden layer does not convey any semantic meaning and the mapping from the intermediate representation to semantically meaningful representation is well-defined, whereas the inverse is not straightforward. This paper is interested in invertible representation learning that maps intermediate representation to semantically meaningful concepts and vice verse. Esser et al. (2020) developed such approach to factorize the hidden representation from a model into user-defined semantic concepts. Specifically, they map arbitrary representations into a space of interpretable representations – a non-linear mapping between the two domains. This mapping is invertible, i.e., any modification in the domain of semantic concepts concurrently alters the original representation. In this scenario, it takes the flattened version of $E(x)$, denoted $z \in \mathbb{R}^N$ (where $N = H \cdot W \cdot C$), and factorize it into $\tilde{z} = (\tilde{z}_k)_{k=0}^K \in \mathbb{R}^N$, where each of the $K + 1$ factors of $\tilde{z}_k \in \mathbb{R}^{N_k}$ with $\sum_{k=0}^K N_k = N$ represents an interpretable concept that is normally distributed $\mathcal{N}(\tilde{z}_k|0, \mathbf{1})$. Calling this transformation $I$, we have $\tilde{z} = I(z)$.

To encode semantic representation into each factor $\tilde{z}_k$, they constrain $(i)$ each factor $\tilde{z}_k$ to vary with exactly one interpretable concept and $(ii)$ $\tilde{z}_k$ to be invariant to all other variations. This is ensured through training pairs $(x^a, x^b)$, which specify semantics through similarity, i.e., image pairs with a semantic concept of *accent chair*. Each semantic concept, indexed by $F \in \{1, ..., K\}$, has image pairs $(x^a, x^b) \sim p(x^a, x^b|F)$ to the corresponding factor $\tilde{z}_F$. To capture the remaining variability that is not captured by the $K$ concepts, a residual concept $\tilde{z}_0$ is introduced. This ensures that the original representation space reflects any change made to the factorized semantic concept $\tilde{z}_k$. Calling this transformation $I^{-1}$, we have $I^{-1}(\tilde{z}) = z$. Intuitively, the goal is to have a bijective mapping so that modifications of the disentangled semantic factors correctly translate back to the original representation. Please refer to Algorithm 2 in Appendix and Esser et al. (2020) for further details.

### 3.2 Concept suppression and transfer

In this section, we focus our attention on controlling the concept transfer by suppressing undesirable concepts in the hidden representation of the source classifier and transferring relevant concepts to the downstream task. Let's continue our running example of classifying "bedroom" vs "living-room" to describe our approach. At the high level, the goal is to take the hidden representation at a layer $L-1$ of the pre-trained model and suppress the *accent-chair* concept without affecting other concepts.

***How to suppress a concept using prototype?*** Let us assume that we have a pre-trained source network $f^s$ that takes input images $x$ from a target task and produces hidden representation at layer $L-1$, $f^s_{L-1}(x) = z^s$, i.e., the layer before classifier head $c^s_L$. Let us define a few semantic concepts specific to the target task as *accent-chair*, *bed*, and *couch*. We first train the CDN $I$ to take the hidden representation $z^s$ and factorize it according to concepts such that $\tilde{z}^s_k = (I(z^s))_k$, where $k \in \{accent - chair, bed, couch\}$. Training is done using pairs of images that contain a common concept, i.e., the same *bed* to map $\tilde{z}^s_{bed}$ or the same *couch* for $\tilde{z}^s_{couch}$. In addition, there is a *residue* factor $\tilde{z}^s_{residue}$ that encodes all other variations unaccounted by these concepts. Since the CDN imposes a one-to-one mapping from the original representation space ($z^s$) to a factorized space ($\tilde{z}^s$), we can edit the factorized representation $\tilde{z}^s_{rotation}$ without affecting the other factors. Given the CDN $I$, suppose one wants to suppress the *accent-chair* concept. We sample a few example images $\{r_1, ..., r_n\}$ which do not have *accent-chair* and pass them through our pre-trained source network and CDN to obtain their *accent-chair* embedding and take the mean to create a *prototype* embedding, $p_{chair} = \frac{1}{n} \sum_{i=1}^n (I(f^s_{L-1}(r_i))_{accent-chair})$, which is indicative of absence of the accent-chair concept. One

---

[1] Where $h, w, c$ are height, width and channel dimensions of input image
[2] Where $H, W, C$ are height, width, and channel dimensions of intermediate representation

can now substitute accent-chair latent embedding of any image using the generated prototype to suppress its presence.

***How to transfer?*** Next, we proceed to training the target model $f^t$ that takes as input images $x$ and maps to a hidden representation at layer $L-1$, $f^t_{L-1}(x) = z^t$, the layer before target task classifier head $c^t_L$. The input image is also passed through the source model to get the source intermediate representation, which is then fed to CDN. The *accent-chair* concept is suppressed by replacing the corresponding factor $\tilde{z}^s_{accent-chair}$ with the *prototype*, $p_{chair}$. The updated hidden representation $I^{-1}(\tilde{z}^s)$ is then transferred to the target classifier $c^t_L$. We consider two variations of transferring knowledge from the source model to the target task,

1. **Controllable Concept Transfer – source** (CCT-s) where we freeze the layers up to $L-1$ of source network, attach a classifier head for target task $c^t_L$ and train the classifier head for the target task with updated source representation, $c^t_L(I^{-1}(\tilde{z}^s))$.

2. **Controllable Concept Transfer – concatenate** (CCT-cat) where we concatenate the updated source representation with that of the pre-trained target network before passing it through the target classifier, $c^t_L([I^{-1}(\tilde{z}^s) \oplus z^t])$ where $\oplus$ represents concatenation operation.

The entire pipeline is presented in Figure 1 and Algorithm 1 as CCT-cat (CCT-s follows similarly but would remove the target network $f^t$ and combination operation in Step 15). Note that this approach works for multiple concepts by generating each *prototype*, $p_i$ and editing the corresponding factorized representation $\tilde{z}^s_i$ for concept $i$. The above descriptions are for when one knows which concepts are to be blocked (in which lines 18-20 of Algorithm 1 should be ignored).

***How to suppress undesirable concepts automatically?*** It may sometimes be the case that there are too many concepts for a human to decide what concepts to transfer/suppress manually. Some concepts may introduce noise to the target task, or one may desire to get insights into which concepts offer performance improvements over the target model. We consider two additional extensions of our framework: 1) **Controllable Concept Transfer – source with concept search** (CCT-s(cs)) and 2) **Controllable Concept Transfer – concatenate with concept search** (CCT-cat(cs)). Both involve a procedure that searches for the optimal subset of concepts (from a predefined set) to suppress (see Algorithm 3 in Appendix). In practice, we learn a Gumbel-softmax distribution for each concept independently to decide whether the concept should be suppressed or not (line 19 of Algorithm 1). We can also use one Gumbel-softmax for all combinations of the concepts, but it may be infeasible when we have a large number of concepts. While we use Gumbel-Softmax sampling, other differentiable methods could also be applied to sample from a structured discrete space (e.g., stochastic softmax (Paulus et al., 2020)).

## 4 How well does Concept Suppression work?

Before we study concept suppression's influence on downstream tasks, we explore how well the concepts are suppressed from source hidden representation. In particular, we consider two sets of experiments. (i) In the quantitative experiments, we adopt mutual information (MI) based metric based on Belghazi et al. (2018) to measure the MI between concept and hidden representation before and after suppressing a concept. In both cases, we edit the model using a single *prototype* to suppress one concept at a time. (ii) In the qualitative experiments, we use an autoencoder architecture as a source model, which facilitates visualization of concept suppression by decoding CDN-edited hidden representations to human-understandable images.

### 4.1 Mutual Information Experiment

This experiment aims to quantitatively assess whether information about undesired concepts is contained or "hiding" in transferred representations. We consider two approaches popular in the interpretable disentanglement of input representations: Invertible Interpretable Network (IIN) (Esser et al., 2020), Concept Activation Vectors (Kim et al., 2018; Zhou et al., 2018; Chen et al., 2020). To formally evaluate the performance of concept suppression in this scenario, we employ an information-theoretic analysis of the transferred

---

**Algorithm 1** Controllable Concept Transfer - Concatenate with Concept Search method (CCT-cat(cs))

---

1: **Inputs:** Target training dataset $D_T$; Target classifier loss $\mathcal{L}_{c^t}(\cdot)$; Combination operation $\bigoplus$; Seed weight parameters: $\mathcal{W}_{c^t}[0]$; Source pre-trained network $f^s$; Target pre-trained network $f^t$; Number of Epochs $E$; Layer $L$; $\mathcal{C}$ List of $K$ Concepts; Number of Epochs $M$ between Concept Searches.
2: $I \leftarrow$ TRAIN-CDN
3: $\tilde{\mathcal{C}} \leftarrow \mathcal{C}$
4: **for** $\mathbb{c} \in \mathcal{C}$ **do**
5:     Randomly sample $n$ images $\{r_1, r_2, ...r_n\}$ *without* concept $\mathbb{c}$.
6:     $p_c \leftarrow \frac{1}{n} \sum_{j=1}^n \left(I(f_L^s(r_j))_{\mathbb{c}}\right)$
7: **end for**
8: Randomly shuffle $D_T$.
9: **for** $epoch \in [1 : E]$ **do**
10:     **for** $batch \in D_T$ **do**
11:         $x \leftarrow D_T[batch]$.
12:         $\tilde{z}^s \leftarrow I(f_L^s(x))$
13:         **for** $\mathbb{c} \in \tilde{\mathcal{C}}$ **do**
14:             $\tilde{z}_{\mathbb{c}}^s \leftarrow p_{\mathbb{c}}$
15:         **end for**
16:         Update $\mathcal{W}_{c^t}[batch] \leftarrow \mathcal{W}_{c^t}[batch-1] - \eta_{batch} \nabla_{\mathcal{W}_{c^t}} \mathcal{L}_{c^t}(f_L^t(x) \bigoplus I^{-1}(\tilde{z}^s))$
17:     **end for**
18:     **if** epoch mod $M == 0$ **then**
19:         $\tilde{\mathcal{C}} \leftarrow$ CONCEPT-SEARCH($\mathcal{C}$)
20:     **end if**
21: **end for**
22: **Output:** Trained model $c_L^t$ with last iterate of $\mathcal{W}_{c^t}$

---

representations. Specifically, we measure the mutual information between a specific concept and the hidden representations. To compute mutual information between concepts and neural network representations, we adapt the *mutual information based neural estimator* (MINE) proposed by Belghazi et al. (2018), where the authors present a way to estimate mutual information between high dimensional random variables using a trainable neural network that they term a *statistics network*. In simple words, given two random variables $C$ for concept and $Z$ for hidden representation,

$$I_\Theta(C, Z) = \sup_{\theta \in \Theta} \mathbb{E}_{\mathbb{P}_{CZ}}[T_\theta] - \log(\mathbb{E}_{\mathbb{P}_C \bigotimes \mathbb{P}_Z}[e^{T_\theta}])$$

where the expectations are estimated using samples drawn from $\mathbb{P}_{CZ}$ and $\mathbb{P}_C \bigotimes \mathbb{P}_Z$ while the objective is maximized by gradient ascent. Please refer to Belghazi et al. (2018) for more details.

In particular, we estimate the MI between the concept $C$ and original hidden representation $z^s$, $I(C, z^s)$, and compare with the MI between concept $C$ and edited hidden representation $\tilde{z}^s$, $I(C, \tilde{z}^s)$. We start with a heterogeneous setup where the source model is trained on the rotated-colored-EMNIST dataset (Cohen et al., 2017), where each colored image is *rotated* by a random angle drawn from $r \in \{90, 180, 270\}$. However, we probe the source model with the rotated-colored-MNIST dataset for MI estimation. Here, MI is measured for *color*, *rotation*, and *digit* for hidden representations – before and after suppressing *color* and *rotation* concepts. To estimate MI, we use the color RGB vector and angle of rotation as concept random variables $C$. These results are presented in Figure 2(a). The MI for *digit* does not change much when suppressing the other two concepts—however, the MI for *color* and *rotation* drop significantly when the respective concepts are suppressed.

We proceed to conduct experiments on the CelebA (Liu et al., 2015) dataset, a large-scale dataset of celebrity faces with attributes. First, a source model is trained on a multi-label classification task to identify four different concepts $\{Smiling, Wearing\_Lipstick, Heavy\_Makeup, High\_Cheekbones\}$. Next, each concept

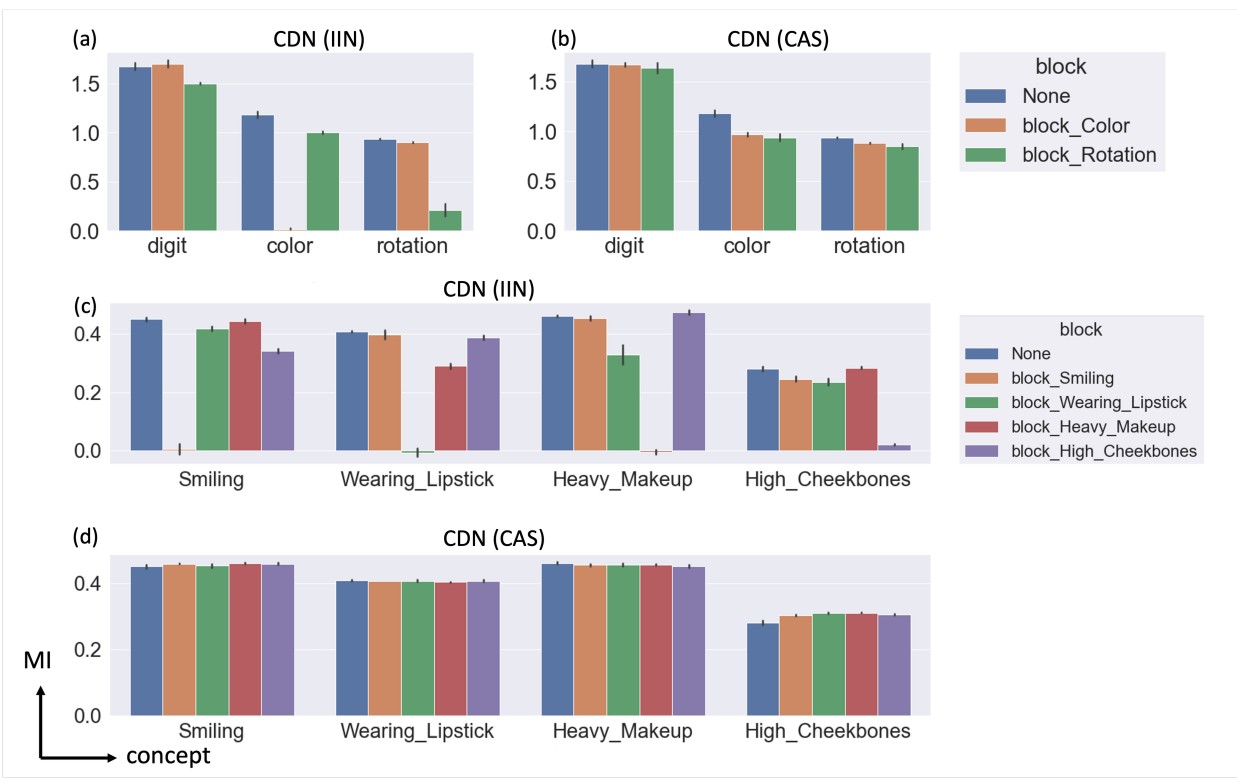

Figure 2: Experiments on mutual information (MI) between concepts and intermediate representation. Each group of bars represents a concept and the color of the bars represents the suppressed concept. MI for rotated-colored-MNIST dataset *digit*, *color*, and *rotation* concepts when *color* and *rotation* are suppressed using (a) CDN (IIN) method and (b) CDN (CAS) method. MI for CelebA dataset where source model is trained on a multi-label classification task to identify 4 different concepts {*Smiling, Wearing_Lipstick, Heavy_Makeup, High_Cheekbones*}. Each concept is then suppressed individually and MI is reported for all concepts using (c) CDN (IIN) method and (d) CDN (CAS) method.

is suppressed, and MI is measured for every concept. To estimate MI, we use the binary value of the presence/absence of each concept as a random variable. Results are presented in Figure 2(b). As demonstrated by the plots, the MI for the suppressed concept reduces (almost) to zero in most cases. These findings suggest that the selected concept is suppressed from the updated hidden representation without significantly affecting the other concepts. Additionally, our choice of suppressing concepts by editing the source's final layers is justified by looking at Figure 8, where we see that more complex concepts appear only in the later layers of the source network.

## 4.2 Comparing against concept activation based suppression (CAS)

To test the need for disentanglement to suppress concepts in hidden representations (IIN), we experiment with concept activation-based suppression (CAS) at the source intermediate representation motivated by previous works (Kim et al., 2018; Zhou et al., 2018; Chen et al., 2020). In this experiment, we train a linear classifier based on logistic regression for each concept. The hidden representation from the source network is used as input, and each model predicts if the concept is present/absent. We then identify the top neurons for each concept based on model coefficients and use them to create the *prototypes*, $p_i$, for each concept. To suppress a concept $i$, we just set those candidate neurons to the corresponding $p_i$. We tested this approach in rotated-colored-MNIST and CelebA by measuring MI values similar to previous experiments. Results are presented in Figure 2(b, d). As demonstrated, the MI for all concepts remains high even after using CAS to suppress them. This suggests that the information about the concept is still present after directly suppressing the hidden representation space. On the other hand, suppressing through the factorized

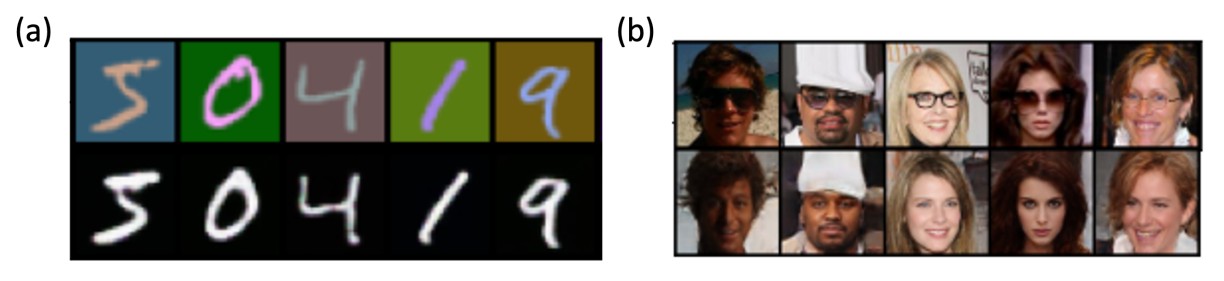

Figure 3: Visualization of concept suppression using an autoencoder as source network. (a) In the first column, we present randomly drawn colored-MNIST images. We then proceed to suppress the concept of *color* from these images and show them in the second row. (b) For CelebA dataset, we consider randomly drawn samples with the *Eyeglasses* attribute and proceed to suppress them. Examples of other concepts being suppressed are in the appendix.

representation of IIN leads to much less mutual information. Hence, we chose to use IIN-based CDN for the remaining experiments.

### 4.3 Qualitative experiments using autoencoder architecture

We replace the source model classifier with an autoencoder to visually evaluate concept suppression. Consider that $f^s$ takes the input image $x \in \mathbb{R}^{h \times w \times c}$ and encodes it into the hidden representation $z = f^s(x) \in \mathbb{R}^N, N = H \cdot W \cdot C$. Decoder $d^s$ is then used to map the hidden representation back to the original image space $\tilde{x} = d^s(z) \in \mathbb{R}^{h \times w \times c}$. First, we consider a simple dataset where we add the concept of *color* to the MNIST dataset (colored-MNIST) (LeCun et al., 1998). A source model is trained to reconstruct the input images, and CDN is trained to disentangle the concepts of *digit* and *color*. We then edit the color factor of a few randomly drawn sample images with the *prototype* $p_c$ of *color* and present results in Figure 3(a). As demonstrated in Figure 3(a), our method can suppress the concept of *color* without affecting the *digit* concept.

Next, we proceed to CelebA dataset. We train the source model to reconstruct CelebA images and then train the CDN to disentangle three concepts – *Eyeglasses*, *No_Beard*, and *Smiling*. We then edit these concepts using corresponding *prototypes*, $p_i$. Results for suppressing the *Eyeglasses* concept in a few randomly drawn sample images are presented in Figure 3(b). Additional figures for suppressing the concept of *Smiling* and *No_Beard* are presented in Figure 9. In Figure 3(b), we see that the concept of *Eyeglasses* is successfully suppressed from sample images, demonstrating that this approach can suppress complicated concepts.

## 5 Evaluation on Classification tasks

The analysis presented in the previous section demonstrates that concept blocking is feasible. In this section, we consider how concept suppression affects the performance of a downstream target task. In particular, we present two scenarios of image classification tasks: (i) Transfer from the rotated-EMNIST trained source model to the rotated-MNIST classification task, and (ii) Transfer between CelebFaces attribute classifiers. For the former task, we use a deep 6-layer convolutional neural network (CNN) for the source model and a shallow 3-layer CNN for the target model. For the CelebFaces task, we used a ResNet34 (He et al., 2016) pre-trained on ImageNet for the source model and ResNet18 for the target model. Additional details about the experimental setup and datasets can be found in Appendix A.3. For each experiment, we consider two variants of transfer: Controllable Concept Transfer – source (CCT-s) and Controllable Concept Transfer – concatenate (CCT-cat) and compare the performance against training the target model independently (TG).

### 5.1 Transfer between heterogeneous datasets

In this experiment, we study the effect of suppressing the concept of *rotation* in a rotated-EMNIST trained source model to a rotated-MNIST transfer task. First, we pre-train the source model with the rotated-

EMNIST dataset, aiming to classify 26 English letters. Next, we train the CDN to factorize the concepts of *digit* and *rotation* using layer $L-1$ representation of the pre-trained source model by probing it with rotated-MNIST images as inputs. Finally, we train the target task of classifying ten digits without suppressing any concepts, termed CCT-s(noedit) and CCT-cat(noedit), and compare the performance to *rotation* suppressed transfer, termed CCT-s(edit) and CCT-cat(edit). To force the target model to rely on the source model for the *rotation* concept, we vary the amount of rotated training samples in the target dataset from 1% to 75%. Top-1 accuracy for these experiments is presented in Table 1 with maximum accuracy in bold for each column. Each experiment is repeated thrice, and the mean accuracy is presented. As expected, at lower percentages of rotated samples ($\leq 10$), we see that suppressing the concept of *rotation* boosts the performance of transfer with CCT-s(edit) and CCT-cat(edit) performing best. We postulate that, at lower percentages of rotated training samples in the target dataset, the target model relies on the source model for a good representation of the *rotation* concept, and suppressing this concept makes the target task easier. As the percentage of rotated samples in the training data increases, the target model learns a better representation of the *rotation* concept and relies less on the source model. This is evidenced by TG and CCT-cat(noedit) having a comparable performance at a higher percentage of rotated samples.

Table 1: Mean accuracy (over three runs) for rotated-EMNIST to rotated-MNIST transfer task. Experiments are conducted by varying the proportion (%) of rotated samples {90, 180, 270} in the training dataset from 1% to 75%. We compare the performance of three models: Target only (TG), CCT-s and CCT-cat. For CCT-s and CCT-cat we conduct experiments without suppressing any concept (noedit) vs suppressing rotation (edit).

| Method | Fraction (%) of rotated images in target task dataset | | | | | | | | | | | | |
|---|---|---|---|---|---|---|---|---|---|---|---|---|---|
| | 1 | 2 | 3 | 4 | 5 | 6 | 7 | 8 | 9 | 10 | 25 | 50 | 75 |
| TG | 0.52 | 0.58 | 0.62 | 0.65 | 0.68 | 0.70 | 0.71 | 0.73 | 0.74 | 0.75 | 0.84 | 0.88 | 0.90 |
| CCT-s(noedit) | 0.57 | 0.59 | 0.61 | 0.62 | 0.63 | 0.64 | 0.65 | 0.66 | 0.67 | 0.68 | 0.75 | 0.80 | 0.81 |
| CCT-cat(noedit) | 0.61 | 0.65 | 0.68 | 0.70 | 0.72 | 0.73 | 0.75 | 0.76 | 0.77 | 0.78 | **0.86** | **0.90** | **0.91** |
| CCT-s(edit) | **0.68** | **0.68** | 0.69 | 0.69 | 0.69 | 0.69 | 0.69 | 0.69 | 0.70 | 0.70 | 0.73 | 0.75 | 0.77 |
| CCT-cat(edit) | 0.64 | 0.68 | **0.71** | **0.73** | **0.74** | **0.76** | **0.76** | **0.77** | **0.78** | **0.79** | 0.85 | 0.89 | 0.90 |

## 5.2 Evaluation on CelebFaces Attributes

In this experiment, we consider a homogeneous setup where the transfer is done between two CelebA tasks. We train the source model to identify four different concepts, *Smiling*, *Wearing_Lipstick*, *Heavy_Makeup*, and *High_Cheekbones*, based on a multi-label classifier. We then train a CDN to factorize these concepts using layer $L-1$ representation of the source network. Finally, we train the target binary gender classification task. Specifically, we train several models with/without suppressing concepts, CCT-s(noedit), CCT-cat(noedit), CCT-s(edit), and CCT-cat(edit), and compare against an independently trained target model (TG).

Table 2: Mean accuracy (over three runs) for CelebA transfer task. Experiments are conducted by suppressing concepts *Smiling*, *Wearing_Lipstick*, *Heavy_Makeup*, *High_Cheekbones* one at a time and suppreesing both *Wearing_Lipstick*, *Heavy_Makeup* at once (*Both_Makeup*). As before we compare the performance of three models: TG, CCT-s, and CCT-cat. For CCT-s and CCT-cat we perform experiments with (edit) and without (noedit) concept suppression.

| Method | Suppressed concept | | | | |
|---|---|---|---|---|---|
| | Smiling | Wearing Lipstick | Heavy Makeup | High Cheekbones | Both Makeup |
| TG | 0.8463 | | | | |
| CCT-s (noedit) | 0.9173 | | | | |
| CCT-s (edit) | 0.9096 | 0.8310 | 0.8493 | 0.9106 | 0.7393 |
| CCT-cat (noedit) | **0.9243** | | | | |
| CCT-cat (edit) | 0.9196 | 0.8746 | 0.8830 | 0.9190 | 0.8353 |

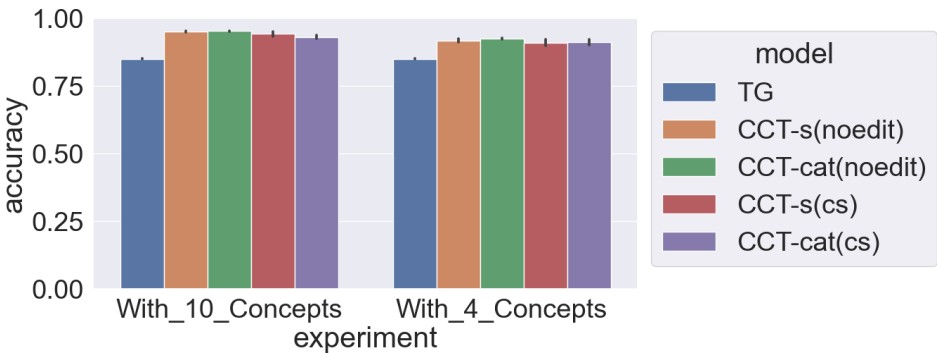

Figure 4: Top-1 accuracy of concept selection methods versus noedit methods in CelebA experiment. Each bar group represents several attributes considered, i.e., four vs ten. Each color represents the model under consideration.

As seen in Table 2, just using the source model CCT-s(noedit) improves performance in comparison to the independently trained target model (TG). We notice that suppressing the concepts of *Smiling* and *High_Cheekbones* did not yield a significant impact on CCT-s(edit) performance, suggesting that these concepts are less relevant to the target task. However, suppressing *Wearing_Lipstick* and *Heavy_Makeup* individually causes a drop in CCT-s(edit) performance. Next, we suppressed both concepts simultaneously and found that the performance dropped further. This suggests that the target classifier is relying on concepts such as *Wearing_Lipstick* and *Heavy_Makeup* for classifying *Male* vs *Not_Male*. We also see that combining source and target representations in CCT-cat(edit) improves the performance compared to CCT-s(edit) while suppressing each concept.

***Concept search based experiments.*** We next add concept search to the same experimental setup and consider additional concepts (up to 10) to account for a situation where the user cannot easily select concepts to suppress themselves. Results are illustrated in Figure 4 using ten concepts (left) and four concepts (right). We see here that concept search can offer similar performance with insights to be gained from examining which concepts are suppressed (most often), which we present in Figure 10. We observe that in 4 concepts experiment, both *Heavy_Makeup* and *Wearing_Lipstick* are suppressed less, and in 10 concepts experiment, *Wearing_Lipstick*, *No_Beard* and *Heavy_Makeup* are top 3 least suppressed concepts.

# 6 Evaluation on co-occurring concepts

Vision datasets often contain concepts that appear together more often than others, and training models using such datasets lead to a phenomenon called *co-occurrence* bias. For instance, a "person" always co-occurs in images with "ski" or "skateboard" (Singh et al., 2020), and hence, a model trained to identify these objects would suffer in the absence of a "person" in them. In this section, we study such scenario where the source dataset has co-occurrence bias and aim to suppress this bias from transferring to a downstream task.

Specifically, it has been shown that object recognition models can spuriously rely on the image background instead of the objects themselves (Ribeiro et al., 2016). We study this phenomenon using the WaterBirds dataset (Sagawa et al., 2019), where waterbirds (or landbirds) are placed against water (or land) more often in training set. In contrast, the test set has an equal number of waterbirds (or landbirds) placed against water (or land). In this experiment, we use an ImageNet pre-trained source model and train our CDN to factorize *Background* and *BirdType*. We then evaluate our downstream waterbirds task with *Background* suppressed source representation and present results in Table 3. We see that suppressing the background information from source representation improves downstream task performance in both CCT-s (edit) and CCT-cat (edit) models, further validating our approach in co-occurring concept tasks.

Table 3: Mean accuracy (over three runs) for WaterBirds transfer task. Experiments are conducted by suppressing *Background* concept. As before we compare the performance of three models: TG, CCT-s, and CCT-cat. For CCT-s and CCT-cat we perform experiments with (edit) and without (noedit) concept suppression.

|                | Accuracy |
|----------------|----------|
| TG             | 0.6592   |
| CCT-s(noedit)  | 0.7812   |
| CCT-cat(noedit)| 0.8459   |
| CCT-s(edit)    | 0.7956   |
| CCT-cat(edit)  | **0.8464** |

## 7    Conclusions

We have seen in this paper how one can effectively suppress certain semantic concepts from being transferred from a source model to a target model while allowing other concepts and information to be transferred. While we were (largely) successful in this endeavor, there may be situations where it is difficult to suppress a certain concept while allowing others to pass. This is because concepts can be (statistically or causally) correlated, so suppressing one will lead to (at least partially) suppressing the other. For instance, it may be impossible to suppress the *Wearing Lipstick* concept while allowing *Heavy Makeup* to be transferred. In the future, it would be interesting to consider cases where the user determines which concepts to suppress, which to pass specifically and to arrive at a strategy that best satisfies these requirements. The strategy may also involve informing the user that the constraints are impossible to satisfy.

Additionally, we suppressed concepts by setting the corresponding latent vector in the IIN disentangled representation to mean/median values based on images lacking that concept and then inverting back to the source intermediate representation. However, there may be other ways to set these values, possibly taking inspiration from the explainable AI and fairness literature (Došilović et al., 2018; Mehrabi et al., 2021), where methods such as SHAP (Lundberg & Lee, 2017) and MACEM (Dhurandhar et al., 2019), there are different ways to determine null/base values indicative of no information.

## 8    Limitations and Future Work

A drawback of our proposed approach is the need for concept attributes (or labels) required to train CDN. However, if the concepts are well-defined, fewer examples could be needed to encode these concepts. When transferring from a pre-trained source model that provides a rich representation to improve the target task, ensuring that these representations are not biased towards a particular concept by suppressing it outweighs the cost of annotating. Additionally, image attribute recognition methods such as (Cheng et al., 2018; Zhang et al., 2022) can be used to identify attributes in a given image as a pre-processing step, but that is out of the scope of this work. An interesting, albeit challenging, direction for future work is learning which concepts are most relevant for transferring to downstream task on the fly.

Most datasets in typical benchmarks used for transfer learning do not provide concept-level attribution, and annotating them is outside the scope of this work. This limited our choice of experimental datasets to those that come with annotated attributes, such as CelebFaces, WaterBirds, etc. We also considered datasets such as MS-COCO, where each image is labelled with person, animal, object, etc., labels. However, we find that the concepts in the CelebFaces dataset are richer in latent concepts. For instance, concepts such as *Heavy_makeup* are more complex, and suppressing them is nontrivial compared to attributes present in MS-COCO, where blocking the pixels of the person bounding box can be considered suppression of the *person* attribute. Although we demonstrated our work on image datasets, one can easily extend our approach to transfer learning setups in tabular and NLP tasks, where each feature can be considered as a concept. We demonstrate such transfer using a proof of concept experiment on a tabular dataset in Appendix A.4. We leave additional exploration of this direction for future work.

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

# A    Appendix

## A.1    Datasets

For the bulk of our experiments in Sections 4 and 5 we use the MNIST (LeCun et al., 1998), EMNIST (Cohen et al., 2017) and CelebFaces Attributes (CelebA) (Liu et al., 2015) datasets. Few examples from each dataset and corresponding concepts are presented in Figure 5

**MNIST**. MNIST is a handwritten digits classification dataset with 10 digits. There are 60,000 training examples and 10,000 testing examples. It is a subset of a larger set from National Institute of Standards and Technology (NIST) where each image is size-normalized and centered in a fixed-size image. We introduce concept of *rotation* to these images by rotating 75% (unless otherwise specified) of images by one of 4 possible values $\{90, 180, 270\}$. Concept of *color* is added to images by multiplying each image channel with corresponding [R,G,B] values drawn from a uniform distribution.

**EMNIST**. EMNIST is a set of handwritten characters derived from NIST Special Database 19. For our experiments we use the EMNIST Letters dataset, which is a 26 class classification of english letters. There are 88,800 training examples and 14,800 testing examples. We introduce concept of *rotation* to these images by rotating 75% (unless otherwise specified) of images by one of 4 possible values $\{90, 180, 270\}$. Concept of *color* is added to images by multiplying each image channel with corresponding [R,G,B] values drawn from a uniform distribution.

**CelebFaces Attributes**. CelebA is a large-scale face attributes prediction dataset with more than 200,000 images. Each image has 40 different attribute annotations. There are 162,770 training examples, 19962 test examples and 19867 validation examples. For our experiments, in Sections 4 we use concepts of $\{Eyeglasses, No\_Beard, Smiling\}$ and for experiments in 5 we use

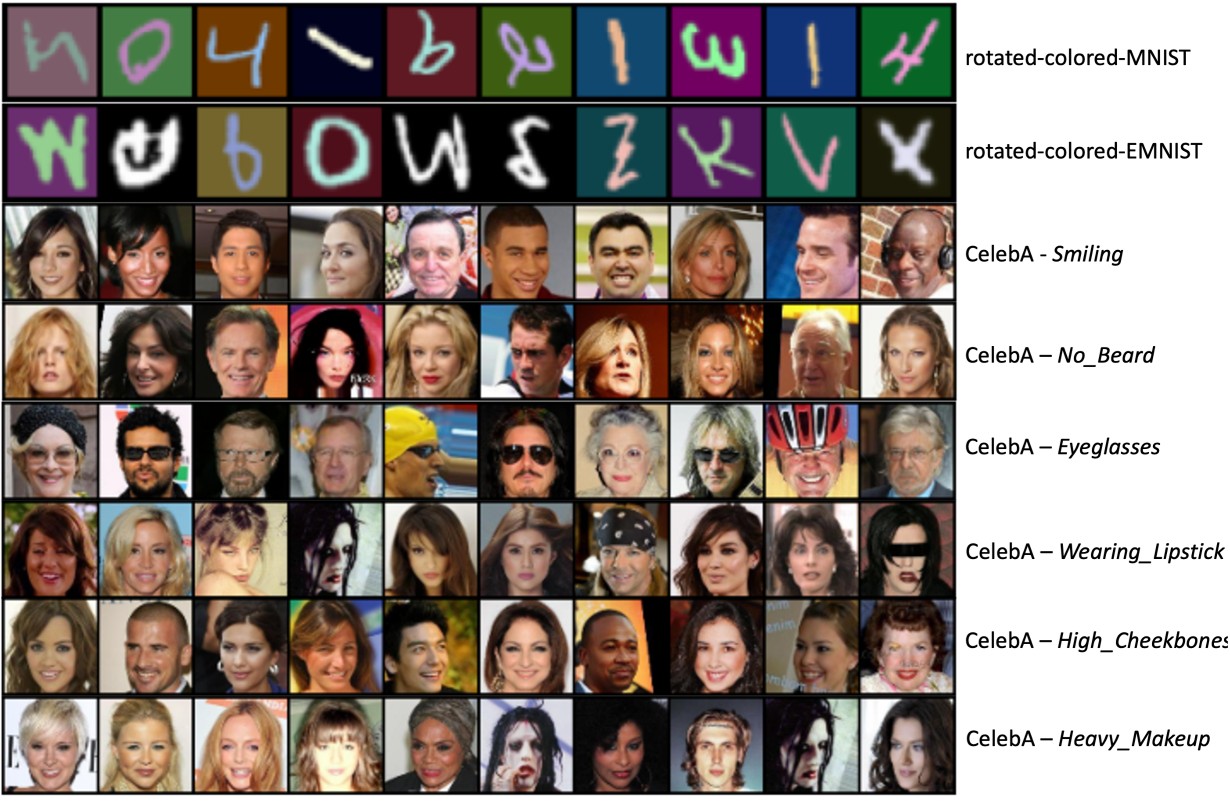

rotated-colored-MNIST

rotated-colored-EMNIST

CelebA - *Smiling*

CelebA – *No_Beard*

CelebA – *Eyeglasses*

CelebA – *Wearing_Lipstick*

CelebA – *High_Cheekbones*

CelebA – *Heavy_Makeup*

Figure 5: Example images for rotated-colored-MNIST, rotated-colored-EMNIST and CelebFaces Attributes.

$\{Smiling, Wearing\_Lipstick, Heavy\_Makeup, High\_Cheekbones\}$. For concept search experiment involving 10 concepts, we chose $\{"Smiling", "Heavy\_Makeup", "Wearing\_Lipstick", "High\_Cheekbones",$ $"Mouth\_Slightly\_Open", "Wearing\_Necklace", "Oval\_Face", "Eyeglasses", "No\_Beard", "Bangs"\}$.

**WaterBirds**. Waterbirds dataset combines bird photographs from the Caltech-UCSD Birds-200-2011 (CUB) dataset (Wah et al., 2011) with image backgrounds from the Places dataset (Zhou et al., 2017). Each bird is labelled as one of Y = waterbird, landbird and place it on one of A = water background, land background, In the training set waterbirds (landbirds) more frequently (90%) appear against a water (land) background, however in the test set there are equal number of samples in each four possible scenarios (waterbird-water, waterbird-land, landbird-water, landbird-land).

## A.2 Models

To describe the architecture used in these experiments, we use the following notation:

- Conv2d($c_{in}$, $c_{out}$, $k$, $s$, $p$): A two dimensional convolution operation that takes $c_{in}$ input channels and produce $c_{out}$ output channels. A square kernel of size $k$ is used. $s$ is the stride and $p$ is the padding.

- ConvT2d($c_{in}$, $c_{out}$, $k$, $s$, $p$): A two dimensional transposed convolution operation that takes $c_{in}$ input channels and produce $c_{out}$ output channels. A square kernel of size $k$ is used. $s$ is the stride and $p$ is the padding.

- Linear($c_{in}, c_{out}$): A linear layer that maps an input vector $v_1 \in \mathbb{R}^{c_{in}}$ to an output vector $v_2 \in \mathbb{R}^{c_{out}}$.

**Autoencoder Architecture**

The architecture for our Autoencoder experiment presented in 4 is based on Lucic et al. (2018). Similar to Esser et al. (2020), we replace the batch normalization (Ioffe & Szegedy, 2015) by activation normalization (Kingma & Dhariwal, 2018). Details regarding the architecture can be found in Table 4.

Table 4: Architecture of autoencoder model. Input images have size of $h \times w \times c$ and are quadratic in nature $h = w$

| Encoder | Decoder |
|---|---|
| Conv2d(3, 64, 4, 2, 1), ActNorm, LeakyReLU(0.2) | ConvT2d($z$, 512, $h/16$, 1, 0), ActNorm, LeakyReLU(0.2) |
| Conv2d(64, 128, 4, 2, 1), ActNorm, LeakyReLU(0.2) | ConvT2d(512, 256, 4, 2, 1), ActNorm, LeakyReLU(0.2) |
| Conv2d(128, 256, 4, 2, 1), ActNorm, LeakyReLU(0.2) | ConvT2d(256, 128, 4, 2, 1), ActNorm, LeakyReLU(0.2) |
| Conv2d(256, 512, 4, 2, 1), ActNorm, LeakyReLU(0.2) | ConvT2d(128, 64, 4, 2, 1), ActNorm, LeakyReLU(0.2) |
| Conv2d(512, $2 \cdot z$, $h/16$, 1, 0) | ConvT2d(64, 3, 4, 2, 1), Tanh |

## MNIST/EMNIST experiments

We use two different architectures for this transfer learning setup. A larger 6 layer network for source model which is the same as the encoder block of auto-encoder presented in Table 4 with a fully connected layer as classifier attached at the end (see Table 5). A smaller 3 layer network for target model (see Table 6).

Table 5: Architecture of source model

| Encoder | Classification Head |
|---|---|
| Conv2d(3, 64, 4, 2, 1), ActNorm, LeakyReLU(0.2) | Linear($z^s$, $n_{classes}$) |
| Conv2d(64, 128, 4, 2, 1), ActNorm, LeakyReLU(0.2) | |
| Conv2d(128, 256, 4, 2, 1), ActNorm, LeakyReLU(0.2) | |
| Conv2d(256, 512, 4, 2, 1), ActNorm, LeakyReLU(0.2) | |
| Conv2d(512, $z^s$, $h/16$, 1, 0) | |

Table 6: Architecture of source model

| Encoder | Classification Head |
|---|---|
| Conv2d(3, 64, 4, 2, 1), ActNorm, LeakyReLU(0.2) | Linear($z^t$, $n_{classes}$) |
| Conv2d(64, $z^t$, $h/16$, 1, 0) | |

## CelebFaces and Waterbirds experiments

We use two different architectures for this transfer learning setup. A larger ResNet34 for source model with a fully connected layer as classifier attached at the end. A smaller ResNet18 network for target model with a fully connected layer attached at the end.

## Concept Distentangling Network

As described in Section 3.1, we use the Invertible Neural Network proposed in Esser et al. (2020) - Invertible Interpretable Network. In this implementation, the network consists of three invertible layers stacked one on to of another to create an invertible *block*: coupling blocks (Dinh et al., 2016), actnorm (Kingma & Dhariwal, 2018) blocks and shuffling blocks. After the input representation $z^s$ is passed through several invertible blocks, the output is split into $K$ factors $(\tilde{z_k^s})_{k=0}^K$. We refer readers to Esser et al. (2020) for further details.

## Algorithm for training CDN

---

**Algorithm 2** TRAIN-CDN: Train Concept Disentangling Network

---

1: **Inputs:** CDN training dataset $D_I$; CDN loss $\mathcal{L}_{\mathcal{I}}(\cdot)$; Seed weight parameters: $\mathcal{W}_I[0]$; Source pre-trained network $f^s$; Number of Epochs $E'$, Layer $L$, $K$ List of Concepts $\mathcal{C}$ to factorize.
2: Randomly shuffle $D_I$.
3: **for** $epoch \in [1 : E']$ **do**
4:     **for** $batch \in D_I$ **do**
5:         $(x_a, x_b | \mathbb{c}) \leftarrow D_I[batch]$. Where $(x_a, x_b)$ are pairs of samples encoding concept $\mathbb{c} \in \mathcal{C}$.
6:         $(z_a^s, z_b^s) \leftarrow (f_L^s(x_a), f_L^s(x_b))$.
7:         $\mathcal{W}_I[batch] \leftarrow \mathcal{W}_I[batch-1] - \eta_{batch,I} \nabla_{\mathcal{W}_I} \mathcal{L}_{\mathcal{I}}((z_a^s, z_b^s) | \mathbb{c})$.
8:     **end for**
9: **end for**
10: **Output:** Trained model $I$ with last iterate of $\mathcal{W}_I$

---

**Algorithm for Concept Search**

---

**Algorithm 3** CONCEPT-SEARCH: Search Concepts to Suppress

---

1: **Inputs:** List of $K$ Concepts $\mathcal{C}$ ; Number of Epochs $E''$; $\mathcal{W}_{cs}$ from previous epoch.
2: Initialize $Gumbel\text{-}Softmax$ weights $\mathcal{W}_{cs}^{\mathbb{c}}$ randomly if not initialized, $\forall \mathbb{c} \in \mathcal{C}$.
3: **for** $epoch \in [1 : E'']$ **do**
4:     **for** $batch \in D_T$ **do**
5:         $\tilde{\mathcal{C}} \leftarrow Gumbel\text{-}Softmax\ (\mathcal{W}_{cs})$.
6:         $x \leftarrow D_T[batch]$.
7:         $\tilde{z}^s \leftarrow I(f_L^s(x))$.
8:         **for** $\mathbb{c} \in \tilde{\mathcal{C}}$ **do**
9:             $\tilde{z}_{\mathbb{c}}^s \leftarrow p_{\mathbb{c}}$
10:         **end for**
11:         $\mathcal{W}_{cs}[batch] \leftarrow \mathcal{W}_{cs}[batch-1] - \eta_{batch,cs} \nabla_{\mathcal{W}_{cs}} \mathcal{L}_{c^t}(f_L^t(x) \bigoplus I^{-1}(\tilde{z}^s))$.
12:     **end for**
13: **end for**
14: **Output:** Selected Concepts $\tilde{\mathcal{C}}$.

---

**Statistics Network**

For mutual information based experiments in Section 4, we adapt a statistics network proposed in Belghazi et al. (2018). We use a custom sequence of 4 layer network that takes intermediate representation $Z$ and concept vector $C$ to estimate the mutual information between them. Details about the architecture is presented in Table 7.

Table 7: Architecture of statistics network for mutual information estimation.

| Statistics Network |
| --- |
| ConcatLayer($Z$,$C$) |
| Linear($|Z| + |C|$, 100), ReLU() |
| Linear(100, 100), ReLU() |
| Linear(100, 1), ReLU() |

### A.3 Experimental details

For our experimental analysis in the main paper, we set the number of epochs for training to $E = 50$ for all models. We train all models using a batch size of 25 and a learning rate of $10^{-4}$ for the Adam optimizer (Kingma & Ba, 2014). All models were randomly initialized before training. While training source classifier, CDN, target classifier, we use different samples to ensure that each model is trained with non-overlapping

samples. For example, we train the source network in CelebA experiments with the standard training dataset, CDN and target classifier with a split of validation samples. Testing is done on the standard test dataset. Statistics network is trained by querying pre-trained source model with standard validation dataset and testing on standard test dataset. For creating prototypes for concepts, we use 100 samples.

The models were trained in parallel with the specifications shown in Table 8.

| Resource | Setting |
|----------|---------|
| CPU | IBM Power 9 CPU @ 3.15GHz |
| Memory | 512GB |
| GPUs | 1 x NVIDIA Tesla V100 16 GB |
| Disk | 1.2 TB |
| OS | RedHat8 |

Table 8: Resources used for training

### A.4 Additional experiments

**Choosing the operation for creating prototypes**

In order to to create prototypes for each concept, we first query the pre-trained source model and CDN with 100 images that don't have a concept, and aggregate the corresponding concept factor in CDN output. To choose the right aggregation factors, we experimented with simple operations such as *mean*, *median*, *mode* and setting the factor to *zero*. In order to visualize how effective each aggregation is at suppressing a concept, we visualize a few images reconstructed by using auto-encoder based source network as detailed in Section 4.1. We present these results for colored-MNIST images with *color* concept suppressed in Figure 6 and CelebA images with *Eyglasses* concept suppressed in Figure 7. We find the *mean* and *median* performs the best and we chose *mean* for rest of the experiments presented in Sections 4 and 5.

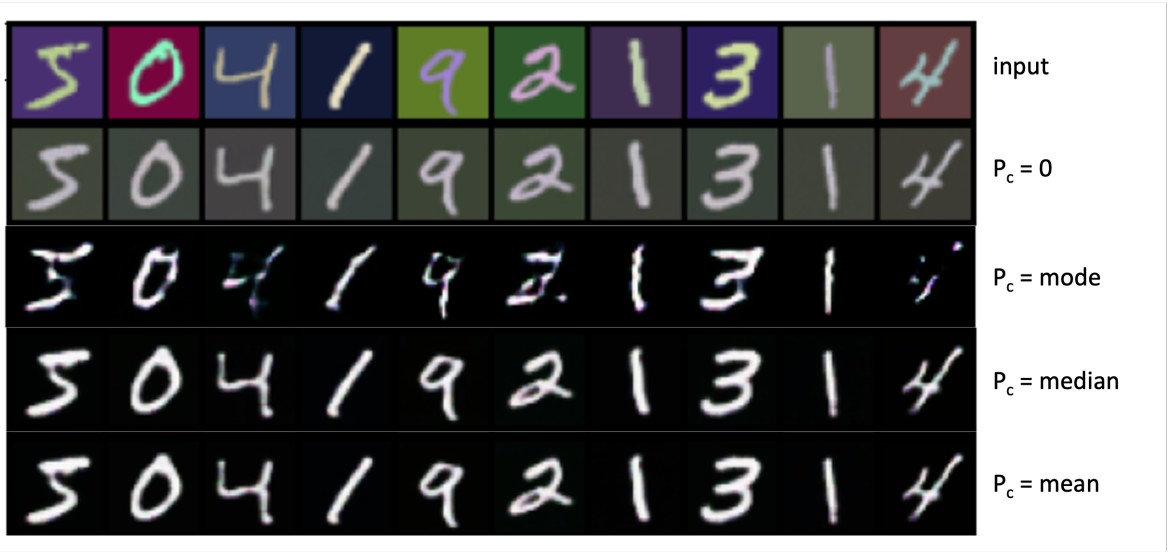

Figure 6: Visualization of *color* concept suppression using prototypes created by different aggregation operators in colored-MNIST images. We find that *median* and *mean* performs the best.

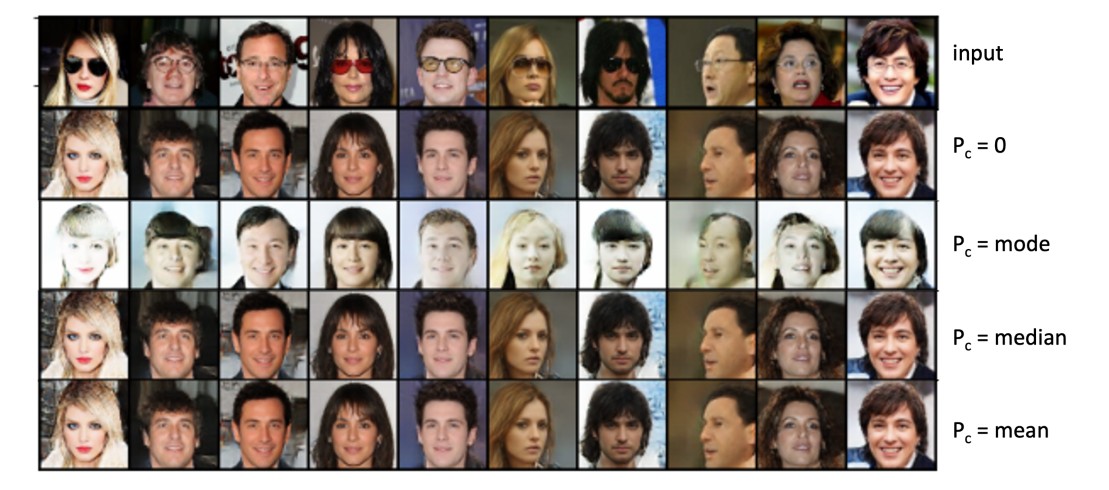

Figure 7: Visualization of *Eyeglasses* concept suppression using prototypes created by different aggregation operators in CelebA images. We find that *median* and *mean* performs the best.

**Concept suppression at different layers of the network**

In this experiment, we compute mutual information for each concept in rotated-colored-MNIST images at different layers of the source network. As demonstrated in Figure 8, we observe that only concept of *color* is learned at layer 2, which is successfully suppressed. More complicated concepts such as *digit* and *rotation* are only learned in later layers. It is thus best to intervene (i.e. suppress) in the final layer (i.e. layer 6), where the model has a good representation of all concepts before transferring the representation to downstream tasks.

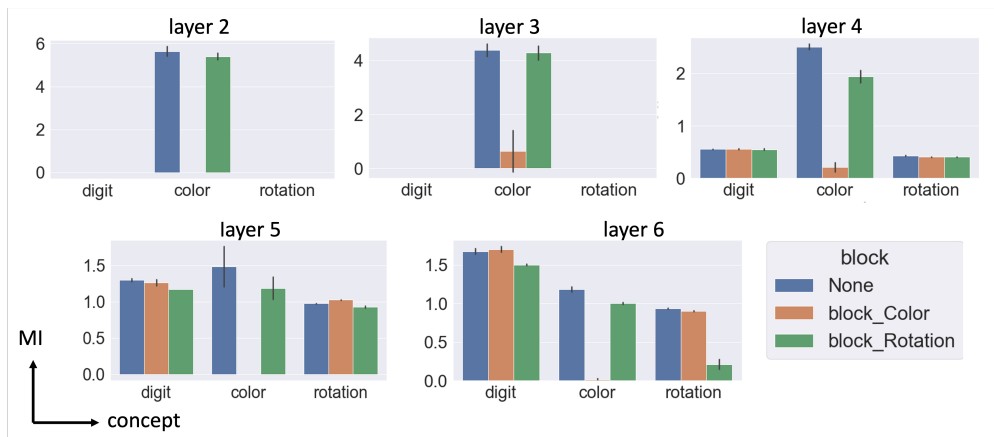

Figure 8: Experiment comparing concept suppressing using hidden representations from different layers for the rotated-colored-MNIST dataset. As can be seen the *color* concept is learned first followed by *digit* and *rotation*. It is thus best to intervene (i.e. suppress) in the final layer (i.e. layer 6) where the model has learned the critical concepts such as *digit* which we want to transfer.

**Autoencoder based concept removal images**

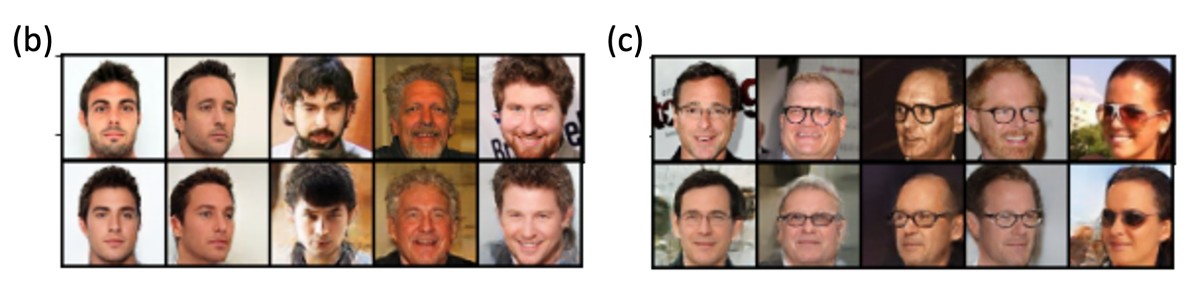

Figure 9: Visualization of concept suppression using an autoencoder as source network. For CelebA dataset, we consider randomly drawn samples with *Beard* & *Smiling* attributes and proceed to suppress them individually. These results are presented in (a) and (b) respectively

**Distribution of suppressed concepts in concept search experiments**

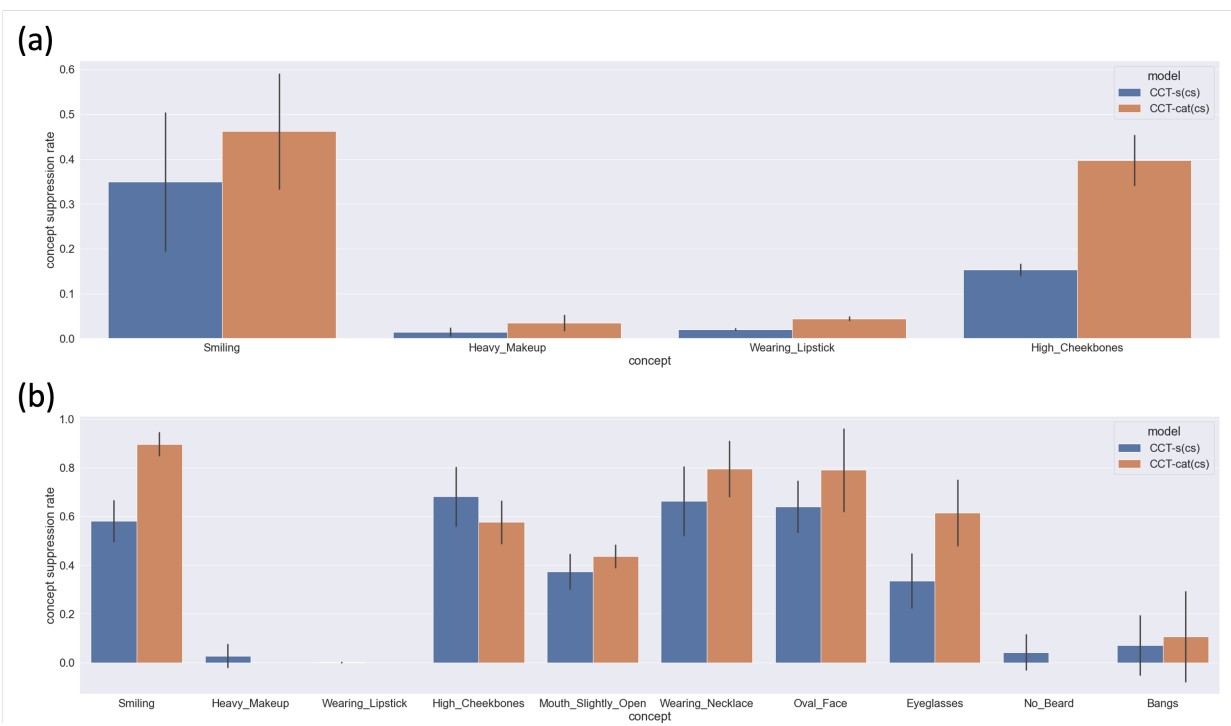

Figure 10: Distribution of concept suppression in the concept search experiments (i.e., CCT-s(cs) and CCT-cat(cs)). We find that the suppressed concepts were meaningful for the task of Male vs. not Male classification, i.e., concepts not related to the task such as *Smiling*, *High_Cheekbones*, and *Oval_Face* were more heavily suppressed. The concept suppression rate (a higher value indicates more suppression) is plotted.

**Experiment using tabular data**

In this section, we discuss a proof of concept experiment using tabular data. Specifically, we consider the CDC Diabetes Health Indicators Dataset (CDC, 2017) available on the UC Irvine Machine Learning Repository. The Diabetes Health Indicators Dataset contains healthcare statistics and lifestyle survey information about people along with their diagnosis of diabetes. This balanced dataset has 70692 samples and we sample 7 features that have high correlation with the diabetes variable.

Table 9: Test accuracy of source and target task using CDC Diabetes Health Indicators dataset.

| Model | Test Accuracy |
|---|---|
| Source | 74.13 ($\pm$0.001) |
| Target (scratch) | 60.70 ($\pm$0.031) |
| Target (fine-tune) | 73.20 ($\pm$0.035) |

We first divide the dataset into source and target tasks, where the source task has 69,692 samples leaving the target task with 1000 samples. For our source task, we train a simple 2 layer multi-layer perceptron (MLP) to diagnose diabetes. Next, we train the target task using a similar 2 layer MLP to diagnose heart disease. Finally, we finetune the pre-trained source model for the target task. Results for this experiment are presented in Table 9. We see that finetuning improves the accuracy of target task in comparison to a target model trained from scratch.

Next, we proceed to edit each independent features by setting the feature values to *zero, random, mean* and *median* values. We present these results in Figure 11. Each bar plot group represents the feature that is removed, whereas the color indicates which operation was used to edit the corresponding feature. Additionally, we plot the Target (scratch) and Target (finetune) results for reference along with their average value plotted as dotted line. We see that even after editing features, accuracy of the scratch model is improved on the target task while remaining lower than the accuracy of the finetuned model without editing. Notably, relevant features such as High Cholesterol, General Health, High Blood Pressure and Age have high impact on heart disease prediction.

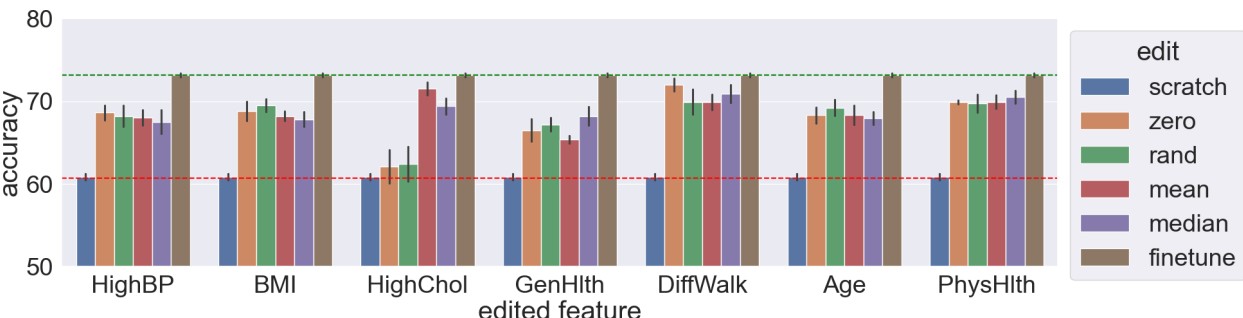

Figure 11: Experimental results for feature suppression on a tabular dataset. Each group of bar plots corresponds to the feature that is edited. The color of each bar represents the operation used for editing. For reference, we add the accuracy of Target(scratch) and Target(finetune) model along with the average accuracy plotted as dotted line.

