# OpenReview forum: "To Transfer or Not to Transfer: Suppressing Concepts from Source Representations"
_TMLR — Accepted by TMLR_

### Review · Reviewer_4k7z · 2023-10-19

**Summary Of Contributions:**

This paper studies a method in order to improve the performance of transfer learning tasks. First, they analyze under some specific settings, a few learned features from the source model can hurt target model accuracy. Then, they propose a pipeline that can remove undesirable information from the source feature vector before transferring it to the target task. To filter out these negative factors, authors utilize an Invertible Interpretable Network (IIN) to disentangle the extracted vector from the source model into a list of concepts. After that, they suppress negative concepts by replacing them with corresponding prototypes (concepts that do not relate to the undesirable concepts). Finally, authors have verified how their approach can improve transfer learning tasks between source and target domains under different experiment settings.

**Audience:**

Yes

**Broader Impact Concerns:**

No concerns about the ethical implications have been identified.

**Claims And Evidence:**

Yes

**Requested Changes:**

Please refer to the "Weaknesses" part of the review.

**Strengths And Weaknesses:**

Below is the list of strengths and weaknesses of this paper, I also include two questions for the authors in the weakness part.

 **Strengths**
* The paper is easy to follow.
* The problem that the authors try to solve is
    important.
* The experimental results have verified clearly the main idea.

**Weaknesses**
* The contribution of this method is limited, your main contribution is just a trick that filters out unnecessary information by replacing it with the prototype before transfer to the target domain.
* Your work may not be the first one in this direction. Under the contrastive learning setting, [1] has proposed a similar method, where their target also removes the factor that has a negative effect when transferred to downstream tasks. I think it could be a good baseline to make a comparison with your method.
* As you have indicated, the main weakness of your method is just to remove the concept that is independent of other ones. In real applications, these concepts tend to have complex correlations with each other. And this unintentionally reduces the impact of this method.
* I think because of the above weakness, your experiment has not shown this approach performance under common benchmarks for transfer learning like CIFAR, ImageNet dataset, etc, which should be considered when working with domain adaptation.
* The writing needs to be polished more, especially the introduction and related works. Some minor problems that I have found: your citation format is not the same throughout the paper, and some of the concept names like the prototype (which is used to replace the undesirable factors), are not highlighted enough about their role.

Besides, I have two questions related to your method:
* How could you define a concept?
* Could you provide the computation cost in order to train these concepts?

**Reference**

[1] Tete Xiao, Xiaolong Wang, Alexei A Efros, Trevor Darrell. "What Should Not Be Contrastive in Contrastive Learning
", ICLR 2021, https://openreview.net/forum?id=CZ8Y3NzuVzO

---

> ### Author Response · Authors · 2023-12-19
> **Rebuttal by authors to reviewer 4k7z**
>
> We thank reviewer 4k7z for helping improve our work and address the concerns raised by the reviewer below,
>
> **Q1. The contribution of this method is limited, your main contribution is just a trick that filters out unnecessary information by replacing it with the prototype before transfer to the target domain.**
>
> A1. We thank the reviewer for raising this concern and providing an opportunity to clarify our contributions. We agree that the main goal is to suppress latent concepts transferred from a source model, either because of regulations (due to bias/fairness, etc.) or because performance could be enhanced by removing unnecessary information. To address the latter, we additionally propose a concept search (cs) variant, where concepts are automatically suppressed to enhance target domain performance. We presented results for this approach in Figure 4. We found that automatically choosing what concepts to suppress by learning a Gumbel weight for each concept leads to similar performance in comparison to the "noedit" variant. As explained in the last paragraph of Section 5.2, concept search offers new insights due to suppressing certain concepts while maintaining performance levels. Additional results for the distribution of suppressed concepts are presented in Figure 10, where the un-suppressed concepts were meaningful for the downstream tasks.
>
> Additionally, for the choice of the prototype itself, we performed qualitative experiments to determine which method performed the best. As demonstrated in Figures 5 and 6, we found that mean and median provided the most realistic suppression of desired concepts. Overall, our approach works on editing arbitrary representation from any given source model, and the edited representation can be transferred to other downstream architectures.
>
> **Q2. Your work may not be the first one in this direction. Under the contrastive learning setting, [1] has proposed a similar method, where their target also removes the factor that has a negative effect when transferred to downstream tasks. I think it could be a good baseline to make a comparison with your method.**
>
> A2. We thank the reviewer for bringing this to our attention, and we agree that the referenced method addressed a similar problem of embedding different semantic concepts in different spaces. To do this, the authors propose to learn several projection functions that take the embedding ($V$) of an image as input and learn embeddings ($V_1, V_2, …, V_N$) such that each embedding space ($V_i$) is invariant to one semantic concept.  Further, they propose two ways to transfer (i) LooC: the general embedding ($V$) or (ii) LooC++: the concatenation of all invariant embeddings {$V_1, V_2, …, V_N$}.
>
> A key differentiator between Xiao et al. and our approach is that we propose to suppress desired semantic concepts while keeping other concepts (largely) intact. On the contrary, LooC/LooC++ can transfer embeddings of one/few/all concepts to a downstream task. This is a limitation, considering that it is impossible to pre-define embedding space for all possible semantic concepts. We have revised the manuscript with this discussion in related work.
>
> **Q3. As you have indicated, the main weakness of your method is just to remove the concept that is independent of other ones. In real applications, these concepts tend to have complex correlations with each other. And this unintentionally reduces the impact of this method.**
>
> A3. Assuming concepts are independent is common in disentangling literature (e.g., Kumar et al., Variational Inference of Disentangled Latent Concepts from Unlabeled Observations, ICLR 2018). However, in practice, there is often a correlation between different concepts. This is seen in the Esser et al., 2020 paper on CelebA as well (Figure 8 there), where adding a beard to a female also changes the female to a male. In our framework, there is flexibility for how suppressed different concepts should be, which offsets the limitation that concepts are not truly independent. As expected, correlated concepts such as Heavy_Makeup and Wearing_Lipstick will remain unsuppressed if they are both useful for the task, as shown in our Figure 10 results, displaying the distribution of concept suppression level from experiments in Figure 4.

---

> > ### Author Response · Authors · 2023-12-19
> > **Rebuttal by authors to reviewer 4k7z**
> >
> > **Q4. I think because of the above weakness, your experiment has not shown this approach performance under common benchmarks for transfer learning like CIFAR, ImageNet dataset, etc, which should be considered when working with domain adaptation.**
> >
> > A4. We thank the reviewer for raising this concern and would like to clarify the choice of datasets in our experiments. The intuition for choosing EMNIST, CelebA, and WaterBirds in our experiments is manifold. Most datasets in typical benchmarks used for transfer learning, such as in https://github.com/thuml/Transfer-Learning-Library, do not provide concept-level attribution, and annotating them is outside the scope of this work. For datasets with attributes such as MS-COCO, where each image is labeled with person, animal, object, etc., labels, we find that the concepts in the CelebFaces dataset are richer in latent concepts. For instance, concepts such as "Heavy_makeup" are more complex, and suppressing them is nontrivial compared to attributes present in MS-COCO, where blocking the pixels of the person bounding box can be considered suppression of the “person” attribute. However, during the rebuttal, we ran additional experiments on the CUB200 bird identification dataset, where we suppressed two latent concepts, the shape and size of birds and one visual concept, primary_color. We verified that suppressing each concept individually had a significant effect on the performance of the source model so that none of them gives a clear story to a transfer learning framework. Additionally, for WaterBirds and CUB200-based target tasks, we use the ImageNet pre-trained source model.
> >
> > **Q5. The writing needs to be polished more, especially the introduction and related works. Some minor problems that I have found: your citation format is not the same throughout the paper, and some of the concept names like the prototype (which is used to replace the undesirable factors), are not highlighted enough about their role.**
> >
> > A5. We thank the reviewer for pointing out inconsistencies in citations. We have addressed these in the revised manuscript. We have updated the text in the introduction and related works to better flow and highlight relevant details, such as prototypes.
> >
> > **Additional questions:**
> >
> > **Q6. How could you define a concept?**
> >
> > A6. We define concepts within the context of images as human interpretable and semantically meaningful attributes. Specifically, we are interested in suppressing latent concepts from the source model to the target model. We view latent concepts as those that cannot simply be removed by masking part of an image. For example, a dog can be removed from an image with masking and replaced with something else. We consider such a concept a visual concept (and this is what we see in Pascal VOC07, MS-COCO, for example). On the other hand, the Heavy_makeup concept from CelebFaces cannot simply be masked from an image and is what we refer to as a latent concept. We have clarified this difference in the introduction of our revised manuscript.
> >
> > **Q7. Could you provide the computation cost in order to train these concepts?**
> >
> > A6. The computational resources used for training are detailed in Appendix section A.3. We used one NVIDIA V100 16GB for all experiments and trained for 50 epochs. Training concept disentanglement using IIN for the WaterBirds dataset (i.e., background and bird type) took an average of 40 GPU **minutes**. And our CelebFaces disentanglement of four concepts (i.e., Eyeglasses, Heavy_Makeup, Wearing_Lipstick, High_Cheekbones) took on average 2 GPU **hours**.

---

### Review · Reviewer_BMpx · 2023-10-24

**Summary Of Contributions:**

This paper studies the transfer of features in out-of-distribution generalization problems. Specifically, the existence of spurious correlation could largely hinder the learning performance on unknown target distributions. To address this issue, the authors propose to suppress some of the harmful features that do not exist in the target domain. Such methodology is instantiated by a concept disentanglement network that leverages the extracted feature of natural images that do not have spurious correlations. By replacing the features from the target domain data with the identified spurious feature that is free from the undesired concept, the newly produced feature can successfully have its concepts suppressed. Therefore, the transfer learning performance could be further improved. Extensive experiments have been conducted to evaluate the performance of this paper.

**Audience:**

Yes

**Broader Impact Concerns:**

The authors have discuss this in the paper, no further concerns.

**Claims And Evidence:**

Yes

**Requested Changes:**

Please follow the weaknesses as a suggestion to improve this paper.

**Strengths And Weaknesses:**

Strengths:
- This paper proposes an interesting idea that can be an effective way to enhance OOD generalization
- The experiments are quite extensive and effective. It is interesting to find that the MI of each concept is successfully suppressed compared to other concepts.

Weaknesses:
- The writing of this paper could be further polished. In my opinion, this paper is not expressed straightforwardly enough.
- Some closely related works that also use the disentanglement technique are not properly referred to:
Mitrovic et al., Representation learning via invariant causal mechanisms, ICLR 2021
Kügelgen et al., Self-Supervised Learning with Data Augmentations Provably Isolates Content from Style, NeurIPS 2021
Shen et al., Weakly supervised disentangled generative causal representation learning, JMLR 2022
Huang et al., Harnessing Out-Of-Distribution Examples via Augmenting Content and Style, ICLR 2023
Zhang et al., Identifiability guarantees for causal disentanglement from soft interventions, arXiv 2023
- The method is still questionable due to three aspects:
	1. How can you be sure that each feature dimension can be specifically assigned to a certain concept? Moreover, how to make sure that the concept features from the prototypes and extracted features are corresponding to each other? If there is no evidence to support the above two questions, this method could be untrustworthy in practice.
	2. This method requires examples to produce prototypes. However, it could be very hard to sample the ideal examples. The authors state that they sample randomly, but the presented examples are nearly identical to their corresponding images in Fig 3.
	3. There is no theoretical analysis to support this concept suppression technique.
- Moreover, why the performance of CAS is much worse than IIN? Is there an intuitive explanation?

---

> ### Author Response · Authors · 2023-12-19
> **Rebuttal by authors to reviewer BMpx**
>
> We thank reviewer BMpx for helping improve our work and address the concerns raised by the reviewer below,
>
> **Q1. The writing of this paper could be further polished. In my opinion, this paper is not expressed straightforwardly enough. Some closely related works that also use the disentanglement technique are not properly referred to: Mitrovic et al., Representation learning via invariant causal mechanisms, ICLR 2021 Kügelgen et al., Self-Supervised Learning with Data Augmentations Provably Isolates Content from Style, NeurIPS 2021 Shen et al., Weakly supervised disentangled generative causal representation learning, JMLR 2022 Huang et al., Harnessing Out-Of-Distribution Examples via Augmenting Content and Style, ICLR 2023 Zhang et al., Identifiability guarantees for causal disentanglement from soft interventions, arXiv 2023**
>
> A1. We updated the text in our paper to flow better and be straightforward to understand while highlighting relevant details. Specifically, we have condensed the introduction while motivating the problem we address. Additionally, we thank the reviewer for pointing to additional disentanglement literature. We added a separate paragraph highlighting various disentanglement literature, including the papers suggested above in Section 2.
>
> **Q2. How can you be sure that each feature dimension can be specifically assigned to a certain concept? Moreover, how to make sure that the concept features from the prototypes and extracted features are corresponding to each other? If there is no evidence to support the above two questions, this method could be untrustworthy in practice.**
>
> A2. We thank the reviewer for these questions and apologize for the confusion. To further clarify, let us consider an example from a waterbird experiment where the goal is to disentangle background and bird-type concepts. First, the image is encoded using a pre-trained ResNet34, resulting in a 512-dimensional embedding (say *C*), which has an entangled representation of desired concepts. Next, we train an invertible interpretable network (IIN) that disentangles 512-dimensional embedding into sets of embeddings that encode for background ($C_{bg}$), bird type ($C_{bt}$), and one embedding that encodes all other variances ($C_{residual}$) such that $C_{bg} + C_{bt} + C_{residual}$ = $C$ = 512. This training is conducted contrastively to enforce maximal similarity between pairs of images that share the same concept (say $C_{bg}$) while penalizing pairs of images with different concepts ($C_{bt}$ and $C_{residual}$). This ensures that each embedding in disentangled space is variant to one and only one pre-defined concept. With this in mind, prototypes can be generated for each concept using the corresponding disentangles embedding vector.
>
> **Q3. This method requires examples to produce prototypes. However, it could be very hard to sample the ideal examples. The authors state that they sample randomly, but the presented examples are nearly identical to their corresponding images in Fig 3.**
>
> A3. We agree with the reviewer that examples are required to generate prototypes, but we would like to clarify that they can be any images that contain the prototype attribute. For instance, to generate a prototype for the concept of $color$, we sample 100 grayscale images that do not have any color in them and pass them through our pre-trained source network and CDN to obtain their $color$ embedding and take its mean. Other operations were also considered, and the results presented in Figures 6 and 7
>
> Additionally, we apologize for the confusion about Figures 3 and 9. We present the original images in the first row, and the images presented in the second row are the same images where the attribute is suppressed using the prototypes. For example, in Figure 3 (a), we take the images in row 1, and for each image, we pass them through our pre-trained source network and CDN and set the $color$ embedding to the prototype of color we outlined above. We then invert the embedding to generate images with suppressed color attribute.
>
> **Q4. why the performance of CAS is much worse than IIN? Is there an intuitive explanation?**
>
> A4. Let us consider the same example from A2. With concept activation-based suppression (CAS), we built a linear classifier for each concept - background and bird type and identified top neurons that fire for the presence/absence of the respective concept. We then created prototypes using these neuronal activations for each concept. Since the representation of different concepts is distributed amongst several neurons, it is possible that the prototypes are ineffective at capturing such non-linear mappings leading to poor suppression.

---

> > ### Comment · Reviewer_BMpx · 2023-12-24
> > **Comment**
> >
> > Thanks for the explanation, I have carefully read your rebuttal and most of my concerns are addressed.

---

### Review · Reviewer_cTKJ · 2023-12-06

**Summary Of Contributions:**

This paper attempts to improve transfer learning by suppressing a specific concept in the intermediate representations of a source model while keeping other concepts (largely) intact. To this effect, two specific algorithms named as CCT-s and CCT-cat are proposed. Evaluations are performed on the classification task across different datasets to demonstrate the effectiveness of the developed framework.

**Audience:**

Yes

**Broader Impact Concerns:**

It would be better if the authors add some short discussions on the social impact side, although I do not see any significant impact concern.

**Claims And Evidence:**

Yes

**Requested Changes:**

I just have some minor exposition issues to point out:

+ It looks that the opening of Section 3 is a bit redundant as the contents are introduced in the **Introduction** Section in a detailed way. So I think the authors could simplify it with a shorter paragraph.
+ I am not an expert in transfer learning. This submission proposes a new methodology to control the intermediate representation of a source model by suppressing a specific concept while keeping other concepts intact, which plays a core role. However, I am a bit confused by why controlling the intermediate representation is better than adjusting the input data. Could the authors append a bit more discussion in the **Introduction** part with some examples? I guess this can make this submission more self-contained?
+ Compared to suppressing concepts in hidden representations (IIN), could the authors provide the classification results on rotated-MNIST dataset (for Table 1) and CelebA dataset (for Table 2) besides Figure 2. This can make the readers have a more comprehensive understanding of the effectiveness of the developed CCT-cat / CCT-s.

**Strengths And Weaknesses:**

**Strengths**

+ The paper is well written with nice figures. I can follow it in an easy way.
+ How to alleviate transfering undesirable information to target tasks is very important to many down-stream applications, which have practical usages.
+ The proposed method is simple. While I am not an expert in this area, this paper seems to be novel to the best of my knowledge.
+ The authors detail their implementation architecture in the appendix, which is very helpful for reproducibility.
+ With the current provided results, the empirical significance can be revealed.


**Weakness**

My concerns mainly lie in the evaluation part:

+ Although it might be unnecessary, I wonder why the authors do not evaluate on the ImageNet dataset, which is bigger and more challenging than the MNIST & CelebA.
+ According to Table 1-3, the quantitative evaluations are performed under the classification tasks. It might be interesting to see comparison results on more challenging tasks like segmentation and generation.
+ It seems that the authors do not compare with other transfer learning methods which rely on fine-tuning and hidden representation matching. Can the authors add some discussions for this? Additionally, the chosen baselines in Section 4.2 are a bit outdated, which were published before 2020.
+ The authors exclusively utilize datasets featuring a limited classification number for their comparisons. What about the results when extending the analysis to encompass 100 categories?
+ In some cases (e.g. Table 1), the results of CCT-cat (noedit) are quite similar to those of CCT-cat (edit). Plus that CCT-cat (noedit) constantly outperforms CCT-cat (edit) in Table 2, this makes me somewhat hesitate to accept the design generality of the CCT-cat framework.

---

> ### Author Response · Authors · 2023-12-19
> **Rebuttal by authors to reviewer cTKJ**
>
> We thank reviewer cTKJ for helping improve our work and address the concerns raised by the reviewer below,
>
> **Q1. Although it might be unnecessary, I wonder why the authors do not evaluate on the ImageNet dataset, which is bigger and more challenging than the MNIST & CelebA**
>
> A1. We thank the reviewer for raising this concern and would like to clarify the choice of datasets in our experiments. The intuition for choosing EMNIST, CelebA, and WaterBirds in our experiments is manyfold. (a) Most datasets in typical benchmarks used for transfer learning, such as in https://github.com/thuml/Transfer-Learning-Library, do not provide concept-level attribution, and annotating them is outside the scope of this work. (b) For datasets that do have attributes such as MS-COCO, where each image is labeled with person, animal, object, etc., labels, we find that the concepts in the CelebFaces dataset are richer in latent concepts. For instance, concepts such as "Heavy_makeup" are more complex, and suppressing them is nontrivial compared to attributes present in MS-COCO, where blocking the pixels of the person bounding box can be considered suppression of the "person" attribute. However, during the rebuttal, we ran additional experiments on the CUB200 bird identification dataset, where we suppressed two latent concepts shape and size of birds, and one visual concept, primary_color, and verified that suppressing each concept individually had a significant effect on the performance of the source model so that none of them gives a clear story to a transfer framework. Additionally, for WaterBirds and CUB200-based target tasks, we use the ImageNet pre-trained source model.
>
> **Q2. According to Table 1-3, the quantitative evaluations are performed under the classification tasks. It might be interesting to see comparison results on more challenging tasks like segmentation and generation.**
>
> A2. The intuition behind choosing classification tasks was driven by the choice of models/datasets chosen for evaluation. However, we do provide image generation results in Figures 3 and 9, where we generate images by blocking individual concepts.
>
> **Q3. It seems that the authors do not compare with other transfer learning methods which rely on fine-tuning and hidden representation matching. Can the authors add some discussions for this? Additionally, the chosen baselines in Section 4.2 are a bit outdated, which were published before 2020**
>
> A3. We thank the reviewer for raising this concern and would like to clarify that the main contribution of our work is a generalized approach to suppress pre-defined concepts from the hidden representation of any pre-trained source model. We demonstrate the effectiveness of downstream tasks using fine-tuning (CCT-s) and representation-concatenation (CCT-cat). Comparing to other vanilla transfer learning methods, which do not aim to suppress undesirable information, is out of scope of this study and will be explored in future work.
>
> With regards to the baselines chosen for our experiment, we gladly invite additional literature to be brought to our attention, but to the best of our knowledge, our current related works section contains the most relevant works for transfer between two general architectures where we want to **suppress** latent concepts. However, we have added additional literature for disentangled representations in related work.
>
> **Q4. The authors exclusively utilize datasets featuring a limited classification number for their comparisons. What about the results when extending the analysis to encompass 100 categories?**
>
> A4. During rebuttal, we performed an additional experiment on the CUB-200 dataset, which aimed to classify 200 different bird species. We suppressed two latent concepts, the $shape$ and $size$ of birds, and one visual concept, $primary-color$. We verified that suppressing each concept individually significantly affected the performance of downstream tasks, suggesting that our approach can be extended to tasks with more than 100 labels.
>
> **Q5. In some cases (e.g.Table 1), the results of CCT-cat (noedit) are quite similar to those of CCT-cat (edit). Plus that CCT-cat (noedit) constantly outperforms CCT-cat (edit) in Table 2, this makes me somewhat hesitate to accept the design generality of the CCT-cat framework.**
>
> A5. CCT-cat was motivated by other transfer learning setups, such as [A], where the representation from the source network is combined with the target model representation. Our goal was to show that even after suppressing concepts in the source representation, CCT-cat (edit), which combines edited source and target representation, improves target task accuracy while guaranteeing that unwarranted concepts are not transferred from the **source model**. By design, CCT-cat (noedit) will have higher/comparable accuracy because the source representation is not edited.
>
> [A] Murugesan, Keerthiram et al. "Auto-Transfer: Learning to Route Transferrable Representations."ICLR2022.

---

> > ### Author Response · Authors · 2023-12-19
> > **Rebuttal by authors to reviewer cTKJ**
> >
> > **Q6. It looks that the opening of Section 3 is a bit redundant as the contents are introduced in the Introduction Section in a detailed way. So I think the authors could simplify it with a shorter paragraph.**
> >
> > A6. We appreciate the reviewer's valuable suggestions. We have updated our manuscripts by condensing the introduction. We have updated section 3's first paragraph for clarity but decided to retain the motivating example since we use the same running example in later paragraphs.
> >
> > **Q7. I am not an expert in transfer learning. This submission proposes a new methodology to control the intermediate representation of a source model by suppressing a specific concept while keeping other concepts intact, which plays a core role. However, I am a bit confused by why controlling the intermediate representation is better than adjusting the input data. Could the authors append a bit more discussion in the Introduction part with some examples? I guess this can make this submission more self-contained?**
> >
> > A7. We briefly talk about this concern in paragraph 3 but agree with the reviewer that it needed further polishing. In this work, we are interested in suppressing latent concepts from the source model to the target model. We view latent concepts as those that cannot simply be removed by masking part of an image. For example, a $dog$ can be removed from an image with masking and replaced with something else. We consider such a concept a $visual-concept$ (and this is what we see in Pascal VOC07, MS-COCO, for example). On the other hand, the $Heavy-makeup$ concept from CelebFaces cannot simply be masked from an image and is what we refer to as a $latent-concept$.
> >
> > Additionally, large pre-trained (source) models are learned with imbalanced/biased data, and retraining these models to remove undesirable concepts might not be ideal. For one, it is very expensive to retrain the source model, and many times, one does not have access to the data used in the source model pre-training. Secondly, retraining with a balanced dataset does not guarantee the removal of a certain concept. Finally, one would need to collect a balanced dataset to account for all different kinds of undesirable information, which can be tedious. We have clarified these details in the updated introduction.
> >
> > **Q8. Compared to suppressing concepts in hidden representations (IIN), could the authors provide the classification results on rotated-MNIST dataset (for Table 1) and CelebA dataset (for Table 2) besides Figure 2. This can make the readers have a more comprehensive understanding of the effectiveness of the developed CCT-cat / CCT-s.**
> >
> > A8. We apologize for the confusion, but the results presented in the first row of both tables titled **TG** are target models trained on rotated MNIST and CelebA without any concept suppression.

---

> > > ### Comment · Reviewer_cTKJ · 2023-12-19
> > > **Thank you for your response**
> > >
> > > Dear Authors,
> > >
> > > Thank you for your detailed reply! Although I still think that it would be better to see the evaluation results on more challenging cases (datasets or tasks) to show the practical usages, your further information addresses my remaining concerns. I am not an expert in this area but I guess persistently chasing the more comprehensive comparisons might be too picky for a research paper. So if other reviewers agree, I keep positive to this submission.

---

### Decision · Action_Editor_K48b · 2024-01-05

**Recommendation:** Accept with minor revision

**Comment:**

This paper investigates the removal of spurious features in transfer learning, with an emphasis on the latent space. Specifically, the nuisance concept is replaced by an *average* concept without considering the nuisance concept.

All the reviewers (including this AE) like the proposed concept and acknowledge the promise of the proposed approach. Simultaneously, several reviewers raised concerns about the paper. Below are the main concerns in the reviews and discussions.

> 1. [About the main contribution]  (This paper) utilized the existing architecture to disentangle intermediate features into the latent spaces with pre-defined concepts ….. Basically, … this contribution is not novel enough.

This AE agrees with the novelty evaluation. However, based on the TMLR acceptance criteria, novelty should not be considered a reason to reject the paper (it may be a factor in considering a featured paper).

Instead, TMLR is based on the following question – would some people in the TMLR audience be interested in the results of this paper? I think the answer is **YES** because all the reviewers think this paper is interesting.


> 2. [About the definition of concept] At this time, these concepts are defined manually or have been pre-defined before (like the CelebFaces dataset), which is hard to obtain and time-consuming in real-world datasets and these concepts could be changed along with tasks….

I think this is a major limitation of the proposed approach. However, in many cases, we have the concept, which is referred to as metadata. Furthermore, I think that the identification (or discovery) of concepts should be considered a new problem in transfer learning. It is very similar to causal discovery. Therefore, it is a bit unrealistic to achieve all objectives within one paper.

> 3.  [About the experiments] its experiment parts can be further improved on large-scale datasets like ImageNet.

I agree with the experimental limitation on high-dimensional datasets such as images. While I think this method has value, for example, considering transfer learning in a tabular or NLP dataset, this method should work because each feature is a concept of its own.


**Decision**

Based on the reviews and my reading, I think this paper is acceptable with minor revisions.
- I would suggest that the authors include a section discussing the limitations or weaknesses (2-3) and possible future actions.
- Regarding weakness (3), I would suggest a proof-of-concept experiment on tabular data. Specifically, it may not be necessary to consider representation learning, but only the operation on the concepts.

**Audience:**

Yes

**Claims And Evidence:**

Yes. Some parts need revisions.